# Deletion of SERF2 in mice delays embryonic development and alters amyloid deposit structure in the brain

Esther Stroo[1],*, Leen Janssen[1],* , Olga Sin[1,2], Wytse Hogewerf[1], Mirjam Koster[3], Liesbeth Harkema[3], Sameh A Youssef[3,4], Natalie Beschorner[5], Anouk HG Wolters[6], Bjorn Bakker[1] , Lore Becker[7] , Lilian Garrett[7,8], Susan Marschall[7] , Sabine M Hoelter[7,8,9], Wolfgang Wurst[8,10,11,12] , Helmut Fuchs[7], Valerie Gailus-Durner[7], Martin Hrabe de Angelis[7,13,14] , Amantha Thathiah[15,16] , Floris Foijer[1] , Bart van de Sluis[3], Jan van Deursen[17], Matthias Jucker[5], Alain de Bruin[3,4], Ellen AA Nollen[1]

In age-related neurodegenerative diseases, like Alzheimer's and Parkinson's, disease-specific proteins become aggregation-prone and form amyloid-like deposits. Depletion of SERF proteins ameliorates this toxic process in worm and human cell models for diseases. Whether SERF modifies amyloid pathology in mammalian brain, however, has remained unknown. Here, we generated conditional *Serf2* knockout mice and found that full-body deletion of *Serf2* delayed embryonic development, causing premature birth and perinatal lethality. Brain-specific *Serf2* knockout mice, on the other hand, were viable, and showed no major behavioral or cognitive abnormalities. In a mouse model for amyloid-*β* aggregation, brain depletion of *Serf2* altered the binding of structure-specific amyloid dyes, previously used to distinguish amyloid polymorphisms in the human brain. These results suggest that *Serf2* depletion changed the structure of amyloid deposits, which was further supported by scanning transmission electron microscopy, but further study will be required to confirm this observation. Altogether, our data reveal the pleiotropic functions of SERF2 in embryonic development and in the brain and support the existence of modifying factors of amyloid deposition in mammalian brain, which offer possibilities for polymorphism-based interventions.

## Introduction

Protein aggregation is a pathological hallmark shared by several age-related neurodegenerative diseases, such as Alzheimer's (AD), Parkinson's, and Huntington's diseases. The amyloid-like aggregates that accumulate in each of these diseases are composed of disease-specific proteins, that is, amyloid-beta (A*β*) and tau in AD (Glenner & Wong, 1984; Masters et al, 1985; Grundke-Iqbal et al, 1986a, 1986b) or alpha-synuclein (*α*-Syn) in Parkinson's disease (Spillantini et al, 1997). Although the exact molecular mechanisms underlying the disease pathology remain to be elucidated, genetic evidence indicates that these aggregation-prone proteins play a key role in the disease processes. Mutations altering the production, processing, and folding of these proteins are sufficient to cause these diseases (Citron et al, 1992; Suzuki et al, 1994; Haass et al, 1995; Borchelt et al, 1996; Duff et al, 1996; Conway et al, 1998, 2000; Cooper et al, 1998; Murayama et al, 1999; Narhi et al, 1999; Miravalle et al, 2000; Murrell et al, 2000; Nilsberth et al, 2001; Choi et al, 2004; Tomiyama et al, 2008; Ni et al, 2011; Ghosh et al, 2013, 2014). Mechanisms underlying age-related protein aggregation include protein homeostatic mechanism that appear to decline with age, which are well studied with the underlying idea that boosting these mechanisms could prevent or delay disease (Labbadia & Morimoto, 2015; Boland et al, 2018; Hipp et al, 2019; Alberti & Hyman, 2021). In contrast, little is known about cellular

[1]European Research Institute for the Biology of Ageing, University of Groningen, University Medical Centre Groningen, Groningen, The Netherlands   [2]Graduate Program in Areas of Basic and Applied Biology, Instituto de Ciências Biomédicas Abel Salazar, Universidade do Porto, Porto, Portugal   [3]Department of Biomolecular Health Sciences, Faculty of Veterinary Medicine, Utrecht University, Utrecht, The Netherlands   [4]Department of Pediatrics, Molecular Genetics Section, University of Groningen, University Medical Centre Groningen, Groningen, The Netherlands   [5]Department of Cellular Neurology, Hertie-Institute for Clinical Brain Research, University of Tübingen, Tübingen, Germany   [6]Department of Biomedical Sciences of Cells and Systems, University Medical Centre Groningen, Groningen, The Netherlands   [7]Institute of Experimental Genetics, German Mouse Clinic, Helmholtz Zentrum München, German Research Center for Environmental Health (GmbH), Neuherberg, Germany   [8]Institute of Developmental Genetics, Helmholtz Zentrum München, German Research Center for Environmental Health, Neuherberg, Germany   [9]Technische Universität München, Freising-Weihenstephan, Germany   [10]Chair of Developmental Genetics, TUM School of Life Sciences, Technische Universität München, Freising-Weihenstephan, Germany   [11]Deutsches Institut für Neurodegenerative Erkrankungen (DZNE) Site Munich, Munich, Germany   [12]Munich Cluster for Systems Neurology (SyNergy), Adolf-Butenandt-Institut, Ludwig-Maximilians-Universität München, Munich, Germany   [13]Chair of Experimental Genetics, TUM School of Life Sciences, Technische Universität München, Freising, Germany   [14]German Center for Diabetes Research (DZD), Neuherberg, Germany   [15]VIB Center for the Biology of Disease, KU Leuven Center for Human Genetics, University of Leuven, Leuven, Belgium   [16]Department of Neurobiology, University of Pittsburgh Brain Institute, University of Pittsburgh School of Medicine, Pittsburgh, PA, USA   [17]Mayo Clinic, Rochester, MN, USA

Correspondence: l.l.s.janssen@umcg.nl; e.a.a.nollen@umcg.nl
*Esther Stroo and Leen Janssen contributed equally to this work

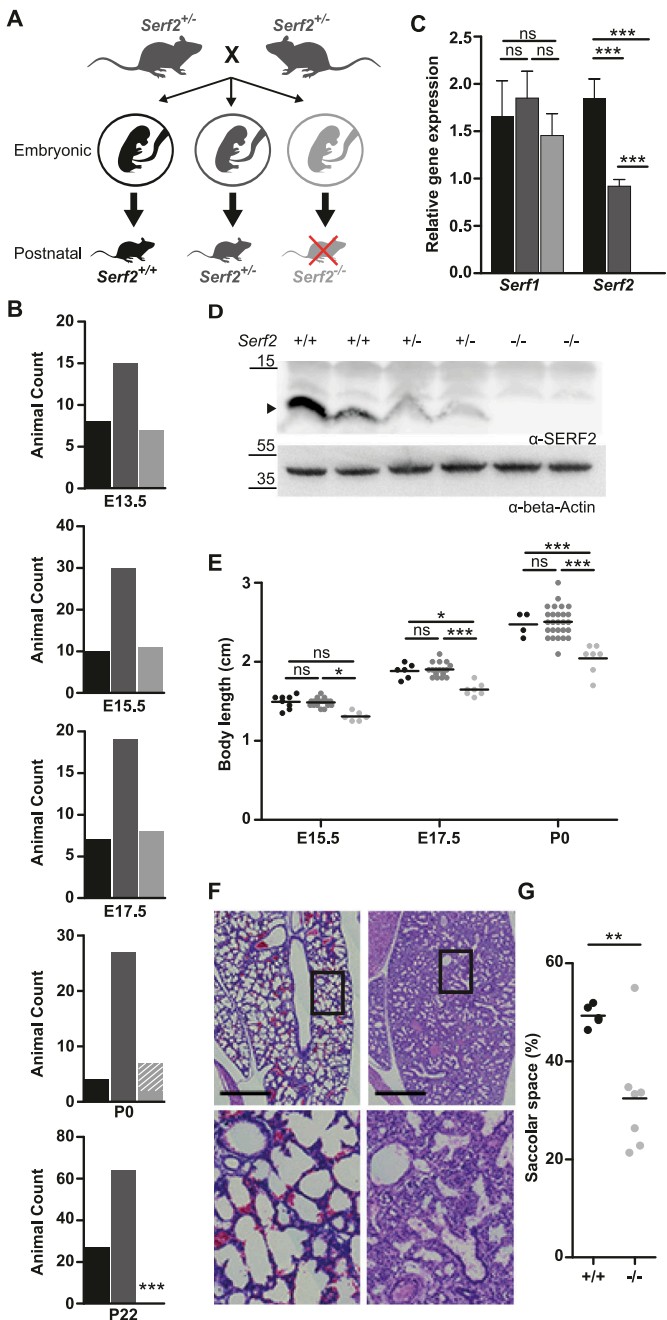

**Figure 1. Full-body *Serf2*$^{-/-}$ mice display a developmental delay and perinatal lethality.**
**(A)** Schematic overview of the crossing scheme with full-body *Serf2*$^{+/-}$ mice and the resulting genotypic composition of the offspring at the embryonic and postnatal stages. **(B)** Absolute animal counts for the three genotypes in the offspring at various days of embryonic development (E13.5, E15.5, and E17.5), at birth (P0) and at time of weaning (P22). **(C)** Real time RT–PCR analyses of *Serf1* and *Serf2* RNA levels in E13.5 heads normalized to housekeeping gene $\beta$-actin (n = 4/group, mean ± SD, one-way ANOVA for each gene, Bonferroni corrected post-hoc comparison between genotypes $^{ns}p_{bon} > 0.05$, ***$P < 0.001$). **(D)** Western blot of SERF2 and actin in *Serf2*$^{+/+}$, *Serf2*$^{+/-}$, *Serf2*$^{-/-}$ embryos at E13.5 (black arrow indicates the SERF2 band). **(E)** Length measurements of *Serf2*$^{+/+}$, *Serf2*$^{+/-}$, *Serf2*$^{-/-}$ embryos at E15.5, E17.5 and P0 (black line = mean, two-way ANOVA with factor age and genotype, Bonferroni corrected post-hoc comparison for *Serf2*$^{+/-}$ and *Serf2*$^{-/-}$ compared with *Serf2*$^{-/-}$ $^{ns}p_{bon} > 0.05$, *$P < 0.05$; ***$P < 0.001$). **(F)** Hematoxylin and eosin stained lung tissue from *Serf2*$^{+/+}$ and *Serf2*$^{-/-}$ pups at P0

mechanisms that drive toxicity and aggregation of aggregation-prone proteins. Inhibiting such driving mechanisms could provide an alternative and complementary approach to treat protein conformational diseases.

Through a genetic screen in *Caenorhabditis elegans* (*C. elegans*), a highly charged cellular protein has been identified that has the capacity to enhance protein toxicity and aggregation of multiple disease-related, aggregation-prone proteins (van Ham et al, 2010). This peptide was dubbed *mo*difier of *ag*gregation-4 (MOAG-4) and was found to be an ortholog of two human proteins: SERF1A and SERF2. Interestingly, SERF proteins accelerate the aggregation of multiple amyloidogenic proteins in vitro, but not non-amyloidogenic proteins (Falsone et al, 2012; Yoshimura et al, 2017; Pras et al, 2021). This aggregation-promoting effect has been accredited to the interaction of a highly positively charged N-terminal segment with segments of the aggregation-prone proteins that are enriched in negatively charged and hydrophobic aromatic amino acids leading to the disruption of its inter and intramolecular electrostatic interactions (Yoshimura et al, 2017; Merle et al, 2019; Meyer et al, 2020; Pras et al, 2021). Neutralizing the charge of MOAG-4 and SERF2 in this N-terminal segment is sufficient to suppress their effect on aggregation and to reduce toxicity in *C. elegans* models for polyglutamine and A$\beta$ pathology (Pras et al, 2021). In this study, we use the established APPPS1-21 mouse model for A$\beta$-pathology to investigate whether the removal of SERF2 also modifies the aggregation of amyloidogenic proteins in the more biologically complex environment of the mammalian brain.

## Results

### Full-body Serf2 KO results in developmental delay and perinatal lethality in mice

To establish the role of SERF2 in vivo, we first ensured SERF2 expression in all major organs, including the brain, and proceeded to generate full-body *Serf2* knockout (KO) mice (Fig S1A–C). During an initial cross, using the four homozygous *Serf2* KO animal (*Serf2*$^{-/-}$) and eight heterozygous *Serf2* KO animals (*Serf2*$^{+/-}$) we obtained, only one *Serf2*$^{-/-}$ survived the day of birth compared with 27 *Serf2*$^{+/-}$ and 96 *Serf2* wild-type (*Serf2*$^{+/+}$) mice (Table S1). After backcrossing into C57BL/6J background, interbreeding of *Serf2*$^{+/-}$ animals did not result in any viable *Serf2*$^{-/-}$ mice at the time of weaning (Fig 1A and B). To determine whether *Serf2*$^{-/-}$ mice died during fetal development, we examined the offspring at multiple developmental stages: embryonic days 13.5, 15.5, and 17.5. In all of the examined embryonic stages, we observed the expected Mendelian ratio of genotypes, around 25% *Serf2*$^{+/+}$ and *Serf2*$^{-/-}$ embryos and around 50% *Serf2*$^{+/-}$ (Chi Square: E13.5 $P = 0.9672$; E15.5 $P = 0.4432$; E17.5 $P = 0.7674$), indicating unaltered viability during embryonic

(left: *Serf2*$^{+/+}$, right: *Serf2*$^{-/-}$, scale bar = 500 $\mu$m, rectangle = 5× magnification in bottom picture). **(G)** Quantification of the saccolar space in lung tissue from *Serf2*$^{+/+}$ and *Serf2*$^{-/-}$ pups at P0 (black line = mean, *t* test **$P < 0.01$). For all mouse data panels: *Serf2*$^{+/+}$ = black, *Serf2*$^{+/-}$ = dark grey, *Serf2*$^{-/-}$ = light grey. Source data are available for this figure.

development (Fig 1B). At birth (P0), however, all $Serf2^{-/-}$ displayed respiratory distress and the majority (5/7) died within 30 min of being born. At the time of weaning, the Mendelian ratio was maintained between the $Serf2^{+/+}$ and $Serf2^{+/-}$ animals, but no $Serf2^{-/-}$ mice survived until this stage (Chi Square: P22 $P < 0.0001$). At E13.5, $Serf2$ mRNA levels showed a 50% reduction for $Serf2^{+/-}$ and complete KO for $Serf2^{-/-}$ compared with $Serf2^{+/+}$ (One-way ANOVA: between all three genotypes $p_{Bon} < 0.0001$) (Fig 1C). No compensatory change in $Serf1$ mRNA levels was observed, suggesting that they function independently from each other (One-way ANOVA: $P = 0.2403$) (Fig 1C). A similar ratio in SERF2 expression was also observed on the protein level (Fig 1D).

Further analysis to uncover the cause of this perinatal lethality revealed an increasing difference in the embryo size of $Serf2^{-/-}$ compared with $Serf2^{+/+}$ and $Serf2^{+/-}$ from E15.5 until birth (Two-way ANOVA body length: E15.5 $Serf2^{-/-}$ versus $Serf2^{+/-}$ $p_{Bon} = 0.0419$, $Serf2^{-/-}$ versus $Serf2^{+/+}$ $p_{Bon} = 0.0874$; E17.5 $Serf2^{-/-}$ versus $Serf2^{+/-}$ $p_{Bon} = 0.0003$, $Serf2^{-/-}$ versus $Serf2^{+/+}$ $p_{Bon} = 0.0157$; P0 $Serf2^{-/-}$ versus $Serf2^{+/-}$ and $Serf2^{+/+}$ $p_{Bon} < 0.0001$) (Fig 1E) (Two-way ANOVA body mass: E17.5 $Serf2^{-/-}$ versus $Serf2^{+/-}$ $p_{Bon} = 0.0124$, $Serf2^{-/-}$ versus $Serf2^{+/+}$ $p_{Bon} = 0.2381$; P0 $Serf2^{-/-}$ versus $Serf2^{+/-}$ and $Serf2^{+/+}$ $p_{Bon} < 0.0001$) (Fig S1D and E). No differences in size could be observed between homozygous wild-type $Serf2^{+/+}$, and heterozygous $Serf2^{+/-}$ embryos at any embryonic stage, including P0

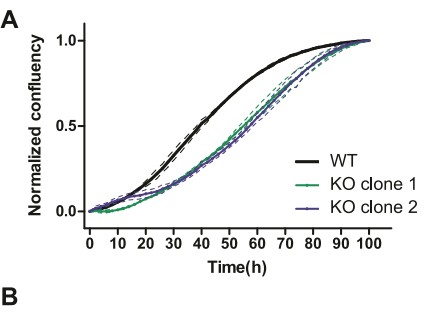

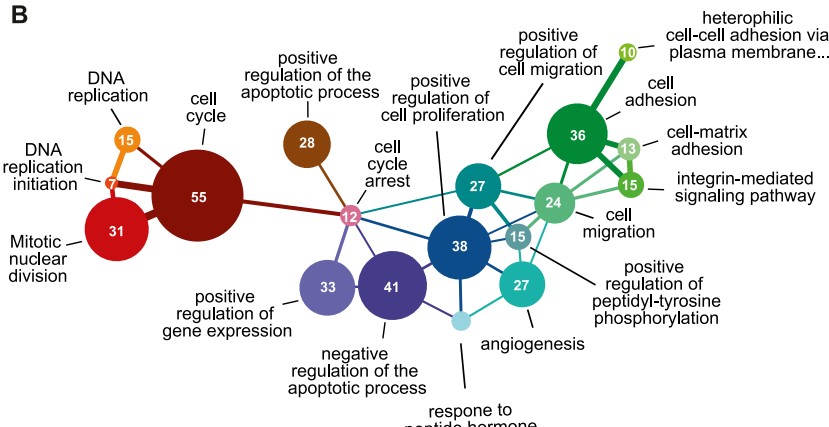

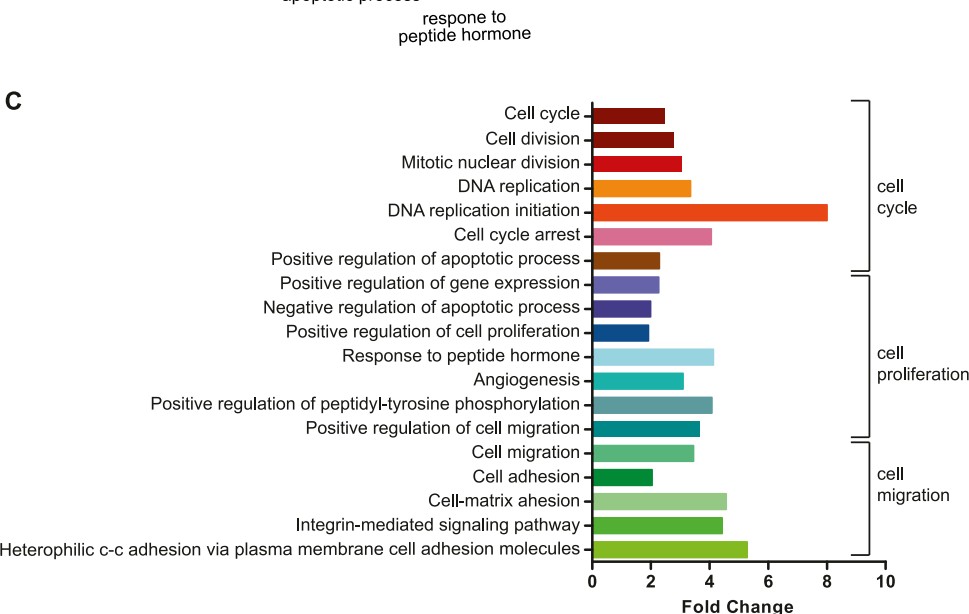

**Figure 2. Cell culture data of *Serf2* KO and control HEK cells and RNA sequencing analysis of MEFs isolated from full-body $Serf2^{-/-}$ and control mice.**
**(A)** Growth curve of two CRISPR-induced $Serf2^{-/-}$ clones of HEK293T versus wild-type HEK293T. (Three replicates measured, full line = replicate average, dashed lines = SD on average). **(B)** Network showing the interconnectivity of enriched GO terms in RNA sequencing data from $Serf2^{-/-}$ and control MEFs. Circle size indicates the amount of differentially expressed genes found in each GO category. Edges were drawn when the minor category shared more than 25% of its found genes with the major category. Edge color indicates the major category in the relationship. Edge thickness indicates the percentage of found genes shared by the minor category. **(C)** Bar chart indicating the fold change for each GO term as determined by DAVID. **(B)** Brackets indicate the three main GO term clusters defined based on the network in (B).
Source data are available for this figure.

(Two-way ANOVA for both body length and weight: at all ages $p_{bon} > 0.9999$). We excluded placental abnormalities, causing an impaired transfer of nutrients and oxygen to the embryo as a cause for the growth delays (Ward et al, 2012). Examination of placenta at all three embryonic stages for microscopic lesions in the placental labyrinth or other changes revealed no genotype-related pathology of the placenta. The results suggest that the absence of SERF2 in the embryo itself was responsible for the observed growth phenotype. Histological examination of various tissues revealed a developmental delay of ~1–2 d in the organs of E17.5 $Serf2^{-/-}$ embryos compared with the previous time points and standard embryonic development (Kaufman, 1992; Kaufman & Bard, 1999). This was most prominent in the lungs, through reduced expansion of alveoli and augmented tissue density (Fig 1F), and in the kidneys, by the increased amount of mesenchyme between the tubules and glomeruli (Fig S1F). At the time of birth, the lungs of $Serf2^{-/-}$ pups displayed a significantly reduced saccular space ($t$ test: $P = 0.0088$) (Fig 1G), reflecting insufficient lung expansion and maturation. This phenomenon, called partial fetal atelectasis, seems to explain the observed respiratory distress and is a likely cause of the perinatal lethality in $Serf2^{-/-}$ pups.

### SERF2 affects growth in a cell-autonomous manner

Given the ubiquitous expression of SERF2 and its overall effect on growth, we examined whether the effect could be cell intrinsic. We generated CRISPR-induced $Serf2^{-/-}$ HEK293T clonal cell lines and found that they also display slower proliferation (Fig 2A). Moreover, RNA sequencing analysis of MEF isolated from E13.5 embryos showed a clear enrichment of GO-terms clustered around three closely related biological functions: cell cycle, cell proliferation, and cell adhesion (Fig 2B and C). Interestingly, in the cell cycle cluster, we observe an overall up-regulation of factors driving cell division and down-regulation of growth arrest factors in

$Serf2^{-/-}$ MEFs (Table S2). The up-regulation of cell cycle regulators may seem counterintuitive in cells that divide slower. However, the up-regulation of cell cycle factor does not automatically mean that cells divide faster. It could be that they need longer time to passage through S or G2 because they experience replication stress and need to repair DNA in the G2 phase. This leads to up-regulation of cell cycle factors as well because they need longer time to passage through these phases. This observation also seems in line with a developmental delay phenotype in $Serf2^{-/-}$ mice, because it is known that cell proliferation is favored over differentiation in earlier stages of embryonic development compared with later stages (Ciemerych & Sicinski, 2005). Overall, the cell culture results point towards a cell-autonomous effect of SERF2 and suggest that deletion of $Serf2$ delays cell growth, which is consistent with a previous report (Tsuboyama et al, 2020). Although our previous study also demonstrated a cell-autonomous function in regulating protein aggregation for MOAG-4, we did not find visible effects of MOAG-4 deletion on the viability or life span of $C.$ $elegans$ (van Ham et al, 2010). Altogether, our results suggest that the perinatal lethality caused by $Serf2$ KO is because of the delayed maturation of certain organs.

### Conditional Serf2 KO mice display a reduction in size, but no other structural abnormalities in the brain

Going back to our initial aim to establish the effect of SERF2 depletion on amyloid aggregation in mouse brain, we needed to circumvent the perinatal lethality of full-body $Serf2$ KO mice. We therefore generated a brain-specific $Serf2$ KO mouse model ($Serf2^{br-/-}$) by combining our $Serf2^{flox/flox}$ mice with the $Sox1$-$Cre$ model (Takashima et al, 2007) (Fig 3A). Brain-specific KO of Serf2 were indeed viable (Chi Square: $P = 0.37$) (Table S3). Analysis of $Serf2$ expression on the mRNA in various organs and on the protein level confirmed brain-specific ablation of $Serf2$ by $Sox1$-mediated $Cre$

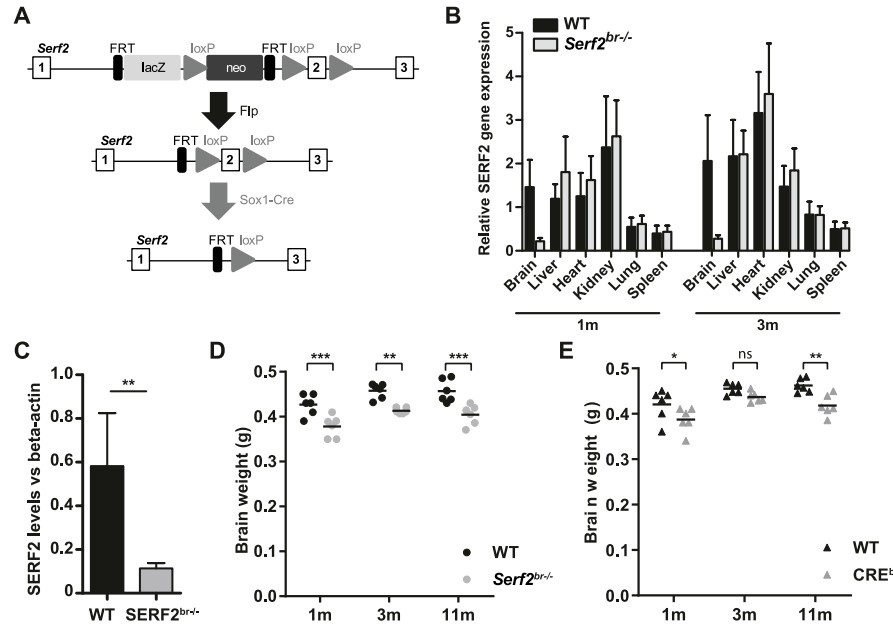

**Figure 3. Brain-specific conditional KO results in reduced brain weight, partially attributed to Cre-expression.**
**(A)** Modified targeting strategy used to delete the exon 2 of $Serf2$ specifically in the central nervous system.
**(B)** Real time RT–PCR analyses of $Serf2$ RNA expression in different organs $Serf2^{+/+}$ and $Serf2^{br-/-}$ female mice at 1 and 3 mo of age. $Serf2$ expression was normalized to housekeeping gene $\beta$-actin (all groups n = 6, mean ± SEM). **(C)** Quantification of $Serf2$ and $\beta$-actin Western blot analysis in brain lysates of WT and $Serf2^{br-/-}$ female mice at 1 mo (both groups n = 6, mean ± SEM, $t$ test **$P < 0.01$). **(D)** Evolution of brain weight in WT and $Serf2^{br-/-}$ female mice between 1 and 11 mo of age.
**(E)** Brain weight of unfloxed WT and CRE[br] mice between 1 and 11 mo of age. For panel (D, E): black line = mean, two-way ANOVA with factors such as age and genotype, Bonferroni corrected post-hoc for genotype comparison at each age $^{ns}p_{bon} > 0.05$, *$p_{bon} < 0.05$, **$p_{bon} < 0.01$, ***$p_{bon} < 0.001$.
Source data are available for this figure.

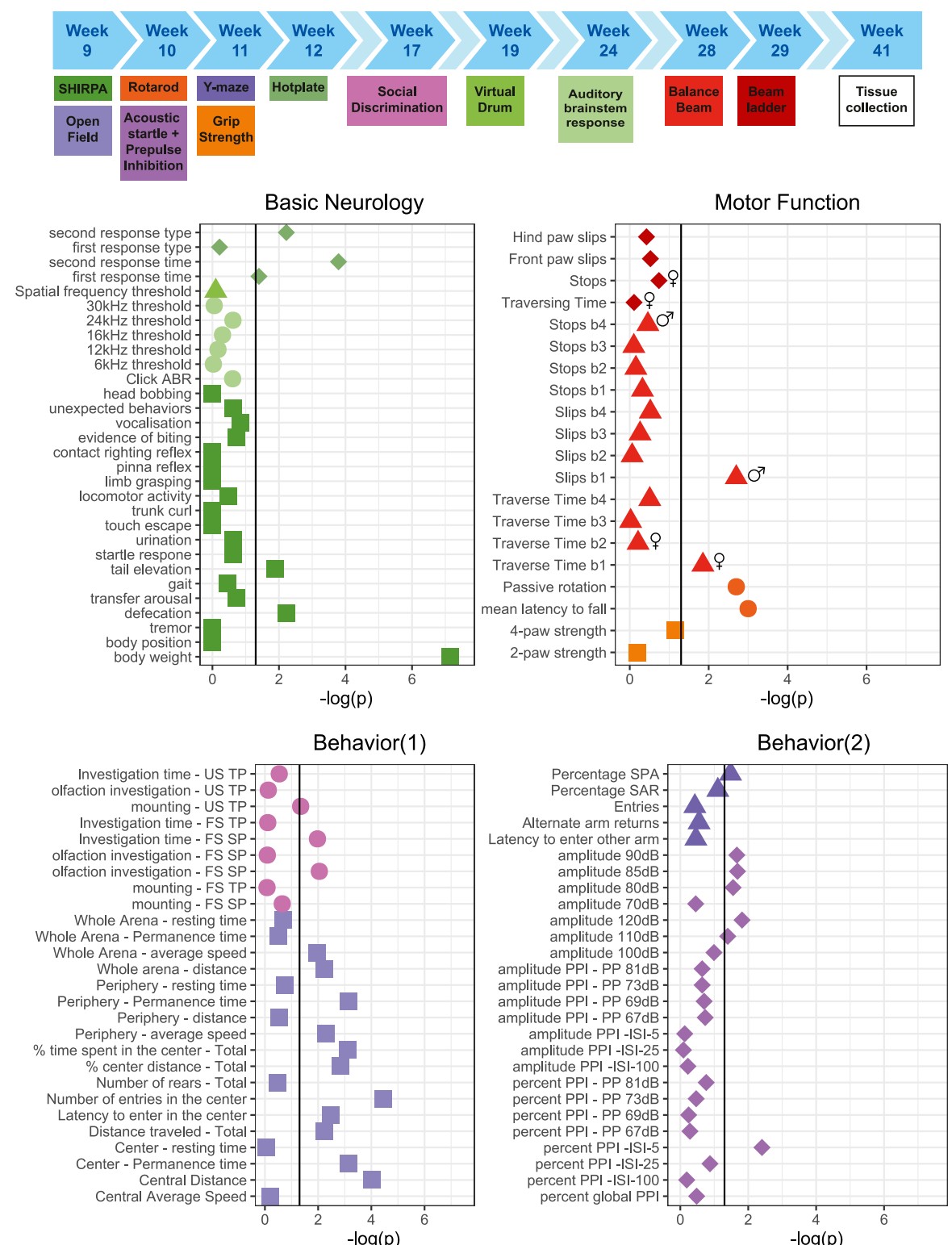

**Figure 4. Extensive phenotyping of conditional *Serf2* KO mice reveals reduced body weight and some subtle behavioral changes, but no clear neurological or motor function defects.**

The top panel provides a schematic overview of the complete phenotyping pipeline used and the ages of testing. The mice were used from a separate breeding line, dedicated to behavioral testing. An overview of the test results is grouped in the four bottom panels according to the three general modalities investigated: basic neurological functioning, overall motor function, and behavior. Graphs show the −log (*P*-value) for the comparison between WT and *Serf2*[br−/−] mice for all parameters tested (vertical black line = significance level of 0.05, sex symbols indicate that sex-specific analysis of the parameter only revealed a significant difference for sex). Source data are available for this figure.

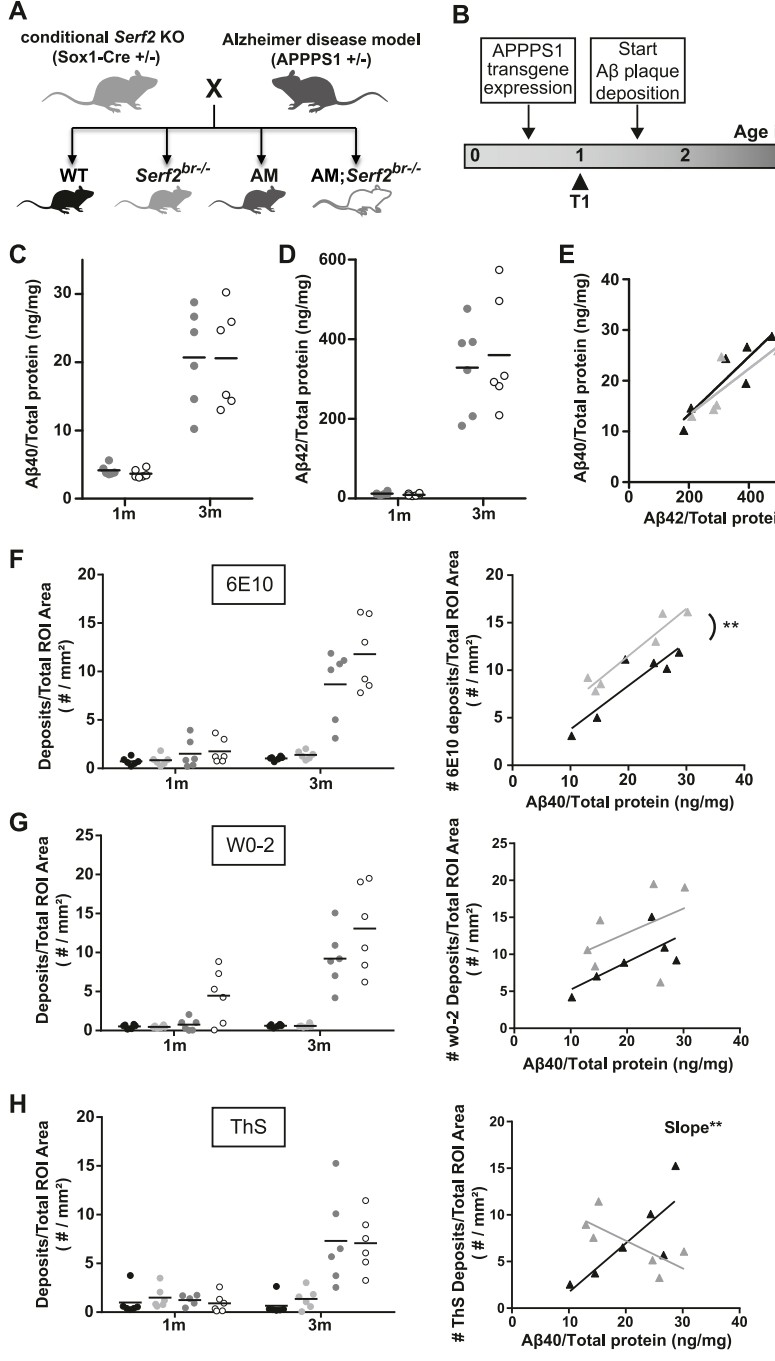

**Figure 5. Crossing a conditional *Serf2* KO with an AM background does not affect Aβ levels, but modifies Aβ plaque formation.**

**(A)** Schematic overview of the cross between brain-specific *Serf2* KO mice and the amyloid model resulting in four experimental groups: WT, *Serf2*$^{br-/-}$, AM, and AM;*Serf2*$^{br-/-}$. **(B)** Timeline for the Aβ pathology in the amyloid model and the selected time points for this study. **(C, D)** Aβ$_{40}$ and (D) Aβ$_{42}$ levels in brain lysate from AM and AM;*Serf2*$^{br-/-}$ mice at 1 and 3 mo of age as determined by ELISA normalized to total protein content. (black line = mean, *t* test between genotypes at both ages not significant). **(E)** Correlation plot depicting the relationship between Aβ$_{40}$ and Aβ$_{42}$ levels in AM (black) and AM;*Serf2*$^{br-/-}$ (grey) mice at 3 mo of age (detailed statistics in Table S4). **(F)** Quantification of the Aβ deposits in the region of interest (ROI) of 6E10 immunostained sagittal brain sections from 1- and 3-mo-old AM and AM;*Serf2*$^{br-/-}$ mice (black line = mean, one-way ANOVA between genotypes at both ages, Bonferroni corrected post-hoc between AM and AM;*Serf2*$^{br-/-}$ not significant). Right panel: correlation plot depicting the relationship between Aβ$_{40}$ levels and 6E10 plaque density in AM (black) and AM;*Serf2*$^{br-/-}$ (grey) mice at 3 mo of age. **(G)** Quantification of the Aβ deposits in the ROI of W0-2 immunostained sagittal brain sections from 1- and 3-mo-old AM and AM;*Serf2*$^{br-/-}$ mice (black line = mean, one-way ANOVA between genotypes at both ages, Bonferroni corrected post-hoc between AM and AM;*Serf2*$^{br-/-}$ not significant). Right panel: correlation plot depicting the relationship between Aβ$_{40}$ levels and W0-2 plaque density in AM (black) and AM;*Serf2*$^{br-/-}$ (grey) mice at 3 mo of age. **(H)** Quantification of the plaque density in the ROI of Thioflavin-S-stained sagittal brain sections from 1- and 3-mo-old AM and AM;*Serf2*$^{br-/-}$ mice (black line = mean, One-way ANOVA between genotypes at both ages, Bonferroni corrected post-hoc between AM and AM;*Serf2*$^{br-/-}$ not significant). Right panel: correlation plot depicting the relationship between Aβ$_{40}$ levels and ThS plaque density in AM (black) and AM;*Serf2*$^{br-/-}$ (grey) mice at 3 mo of age. In all panels, ROI = cortical and hippocampal area; For all dot plots: WT = black, *Serf2*$^{br-/-}$ = light grey, AM = dark grey and AM;*Serf2*$^{br-/-}$ = white; For correlation plots: AM = black and AM;*Serf2*$^{br-/-}$ = grey, detailed statistics in Table S4; \*\*\**P* < 0.001; \*\**P* < 0.01; \**P* < 0.05.
Source data are available for this figure.

expression in our *Serf2*$^{flox/flox}$ mice (*t* test WT versus *Serf2*$^{br-/-}$ brain: 1m *P* = 0.0022; 3m *P* = 0.00527) (Figs 3B and C and S2A). The small traces of Serf2 expression detected in the brains of the KO mice is most likely because of the presence of cells from non-neuronal lineage in the samples, for example, cells from the circulatory system. In correspondence with the findings from the full-body KO, we did observe an overall reduction in the brain weight of *Serf2*$^{br-/-}$ of around 10% compared with WT (two-way ANOVA: genotype *P* < 0.0001) (Fig 3D). This difference was already present at 1 mo of age and remained at least up to the age of 11 mo. Both WT and *Serf2*$^{br-/-}$

displayed similar increases in brain weight between 1 and 3 mo of age because of ongoing brain maturation (two-way ANOVA: interaction *P* = 0.8951). This suggests that the reduction in brain size takes place during an early stage of brain development, but that these changes do not necessarily alter continued development. Additional histological examination of the brains with hematoxylin–eosin staining revealed no difference in the cell density (Fig S2F), nor did we find any evidence of degeneration, apoptosis or necrosis. Apart from the overall reduction in weight, the brains showed no structural abnormalities and the tissue

appeared healthy. We also examined the *Sox1-Cre* mice (Takashima et al, 2007) without floxed *Serf2* (*Cre^br^*) to distinguish between the effects of *Cre* expression and *Serf2*^br−/−^. The *Cre^br^* mice also displayed a reduction in brain weight compared with the WT controls (two-way ANOVA: genotype $P$ = 0.0001) (Fig 3E), but overall, it was less pronounced than in *Serf2*^br−/−^ animals. Therefore, it is possible that the expression of Cre alone contributed to the observed decrease in brain size. Overall, brain deletion of SERF2 did not affect the viability or structural integrity of the brain.

## Brain-specific deletion of *Serf2* show no major neurological defects

Next, we performed extensive phenotyping of the *Serf2*^br−/−^ to investigate basic neurologic functioning, motor function, and behavior (Fig 4). The most prominent difference we observed in our brain-specific deletion model was an overall reduction in body weight compared with WT mice, around 8% for females and 15% for males. In addition, the *Serf2*^br−/−^ mice appeared slightly more agitated as evidenced by the increased tail elevation and defecation in the SHIRPA (Fig 4, basic neurology panel). The *Serf2*^br−/−^ mice were also more likely to refrain from a secondary response in the hot plate test within the test period of 30 s, which could indicate altered or retarded thermoreception. However, we cannot exclude that this phenotype was confounded by alterations in other functions. In the tests for motor functions, *Serf2*^br−/−^ mice show a reduced tendency to passive rotation and reduced latency to falling for both sexes. Although this could indicate a motor deficit, none of the other motor tests consistently demonstrated a motor defect across sexes and measured parameters. Moreover, behavioral differences may also influence rotarod performance and could therefore be the basis of the observed difference. In the behavioral assays, we observed a clear difference in the open field measures for distance travelled and center time, again indicating a hyper-responsiveness to mild novelty stress. In the other behavioral tests, we also observed some mild differences in certain parameters, such as decreased percentage of spontaneous alternations in the Y-maze, reduced social affinity, altered acoustic startle at specific amplitudes, and reduced pre-pulse inhibition, but only at the lowest interstimulus interval. Overall, the latter changes appear less robust and require confirmation in an independent cohort. Although the difference in brain and body size is definitely a factor to be considered during further analysis, our extensive phenotyping of *Serf2*^br−/−^ mice revealed no major deficits in neurological or motor function and only revealed some mild behavioral alterations, predominantly a slight hyperresponsiveness and increased agitation. Therefore, we decided to proceed with these brain-specific knockout mice to investigate the effect of *Serf2*^br−/−^ on Aβ brain pathology.

## Brain-specific deletion of SERF2 in APPPS1 mice preserves APP levels or Aβ production

Conditional *Serf2*^br−/−^ mice were crossed with the *APPPS1-21* amyloid (AM) model (Radde et al, 2006), which contains human transgenes for both APP with the Swedish KM670/671NL mutation and PSEN1 with the L66P mutation. From this cross, we obtained four experimental groups: WT, *Serf2*^br−/−^, AM, and AM;*Serf2*^br−/−^ (Fig 5A). Upon crossing of the *Serf2*^br−/−^ with the amyloid model, we observed the expected Mendelian ratio for all the genotypes at P22, indicating that the viability of AM mice was not affected by brain-specific deletion of *Serf2* (Chi Square: $P$ = 0.37) (Table S3). Based on the known progression of Aβ plaque pathology in the AM mice, we selected two age groups for further analysis: before (1 mo) and after (3 mo) Aβ plaque deposition (Fig 5B). Analysis of the *Serf2* mRNA and protein levels in both age groups showed that *Serf2* expression was not altered by the Aβ pathology (Two-way ANOVA: *Serf2*^br−/−^ and AM; *Serf2*^br−/−^ versus WT 1m $p_{bon}$ < 0.001; 3m $p_{bon}$ < 0.0001) (Fig S2B and C). The *Serf2*^br−/−^ and the AM;*Serf2*^br−/−^ mice again exhibited the reduction in brain weight previously observed with *Serf2*^br−/−^ (two-way ANOVA: *Serf2*^br−/−^ versus WT both ages $p_{bon}$ < 0.01; AM;*Serf2*^br−/−^ versus WT 1m $p_{bon}$ < 0.0001; 3m $p_{bon}$ < 0.001) (Fig S2D). Also, *Cre^br^* mice again showed a similar, albeit slightly less pronounced reduction in brain size (Fig S2E) and hematoxylin–eosin staining revealed no difference in the cell density (Fig S2F) or tissue health because of *Serf2* deletion.

Next, we examined whether the brain KO of *Serf2* affected any of the key components in the Aβ aggregation process, like altering the levels of APP or the production of its cleavage products, Aβ$_{40}$ and Aβ$_{42}$. We determined the expression of human *APP* in WT, *Serf2*^br−/−^, AM, and AM;*Serf2*^br−/−^ mice at both 1 and 3 mo of age and observed no difference because of brain-specific *Serf2* KO ($t$ test: 1m $P$ = 0.9170; 3m $P$ = 0.9963) (Fig S2G, data from WT and *Serf2*^br−/−^ not included because of lack of APP construct and absence of signal). Western blot analysis confirmed there was also no difference in APP protein levels between AM and AM;*Serf2*^br−/−^ ($t$ test: 1m $P$ = 0.1157; 3m $P$ = 0.5908) (Fig S2H and I). Further analysis of Aβ$_{40}$ (two-way ANOVA: genotype $P$ = 0.8841; age $P$ < 0.0001; interaction $P$ = 0.8841) (Fig 5C) and Aβ$_{42}$ (two-way ANOVA: genotype $P$ = 0.7006; age $P$ < 0.0001; interaction $P$ = 0.6522) (Fig 5D) showed an increase in concentration between 1 and 3 mo, but this was the same in both AM and AM; *Serf2*^br−/−^. Given the variability in the Aβ concentrations at 3 mo and the fact that the ratio between these two peptides is known to affect the aggregation process, we also investigated the correlation between these two Aβ levels (Fig 5E). This analysis showed that mice with a high Aβ$_{42}$ concentration display a similarly high Aβ$_{40}$ level, maintaining a comparable Aβ$_{40}$/Aβ$_{42}$ ratio for both AM and AM; *Serf2*^br−/−^ animals. Similar to previous observations with MOAG-4 in *C. elegans*, these data suggest that SERF2 does not affect the levels of APP and its cleavage products, Aβ$_{40}$ and Aβ$_{42}$.

## Serf2 KO alters the amount of Aβ deposits in the brain

We next investigated whether SERF2 affects the Aβ aggregation by performing immunohistological analysis of the Aβ plaque pathology. Initially, we performed a general Aβ staining using the 6E10 antibody directed at the N-terminal amino acid residues 1–16. In accordance with the known progression of Aβ plaque pathology in the amyloid model, we only found Aβ deposits in the 3-mo-old AM and AM;*Serf2*^br−/−^ mice. As expected, most plaques were found in the cortex, with some pathology beginning to spread to the hippocampus as well (Fig S3A). The Aβ deposits displayed a broad range of sizes, but we found no differences in global morphology between the plaque population found in AM

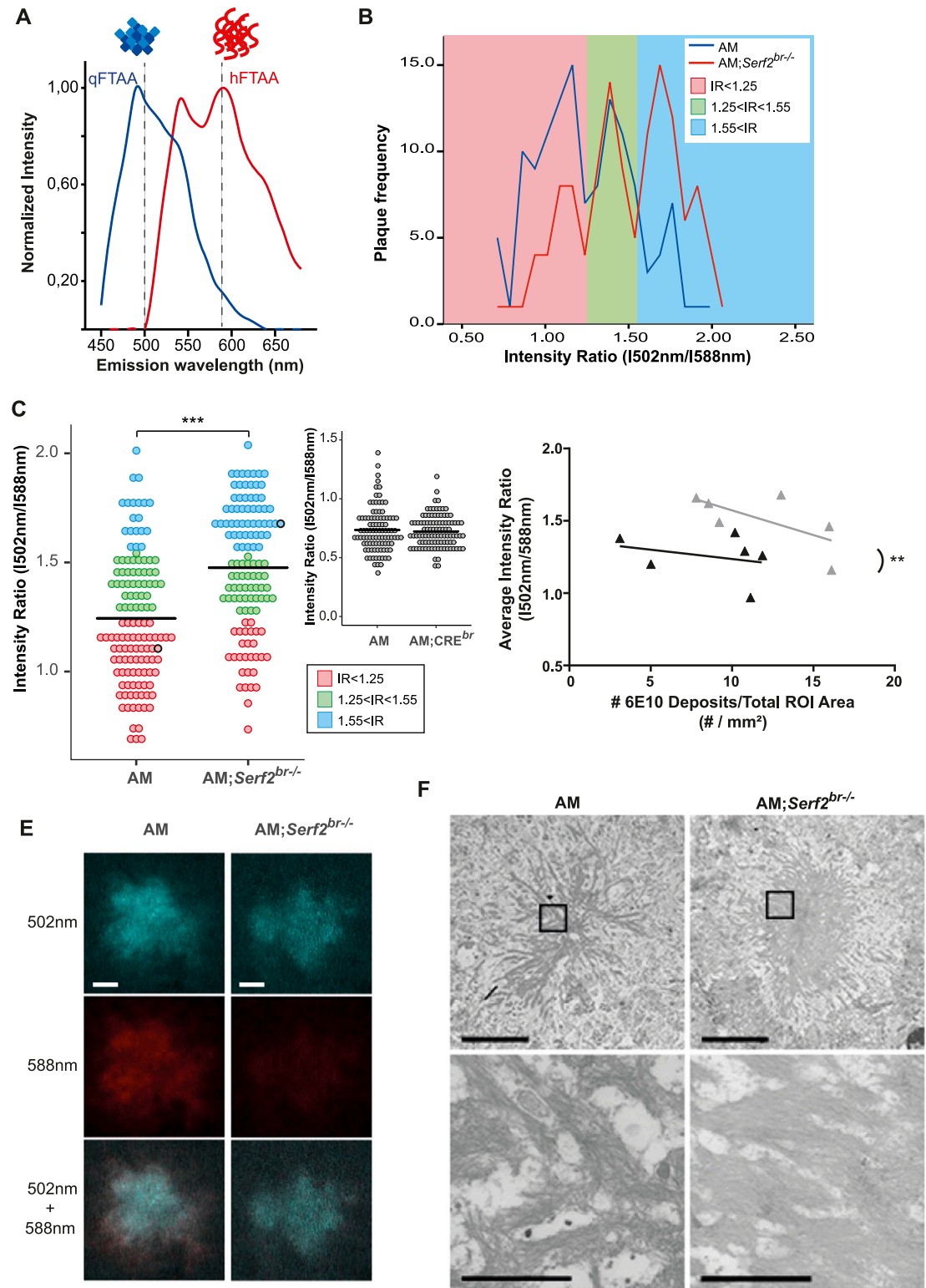

**Figure 6. Brain-specific *Serf2* KO alters the structural composition of amyloid plaques.**
**(A)** Graph showing the theoretical LCO dye emission spectra for qFTAA, which binds mature Aβ fibrils (blue), and hFTAA, which binds both mature fibrils and prefibrillar Aβ species (red). **(B)** Frequency distribution of the ratio in fluorescence intensity at 502 nm and 588 nm for all amyloid plaques identified by confocal microscopy in AM (blue) and AM;*Serf2*$^{br-/-}$ (red) mice at 3 mo of age. Three peak frequency categories were identified: low IR (red), medium IR (green), and high IR (blue) (n = 6 mice/genotype). **(B, C, E)** Dot plot representing the intensity ratio of all amyloid plaques identified by confocal microscopy for AM and AM;*Serf2*$^{br-/-}$ mice at 3 mo of age (colors represent the IR categories defined in panel (B), black circles correspond to the plaques depicted in panel (E), black line = mean, *t* test ***P < 0.001) (n = 6 mice/genotype).

and AM;*Serf2*$^{br-/-}$ (Fig S3B–D). Quantification of the 6E10-positive deposits showed a slight increase in AM;*Serf2*$^{br-/-}$ compared with AM, but this just failed to reach statistical significance in the Bonferroni corrected post-hoc comparison (two-way ANOVA: all 3m non-AM versus AM groups $p_{bon}$ < 0.0001; AM versus AM; *Serf2*$^{br-/-}$ $p_{bon}$ = 0.063) (Fig 5F, left panel). Given the high variability in plaque load between animals within the same experimental group, we examined whether this variability reflected the previously observed variation in A$\beta$ levels. Unsurprisingly, animals with higher levels of A$\beta_{40}$ (Fig 5F, right panel) and A$\beta_{42}$ (Fig S5A) displayed a higher overall plaque load in both groups. More interestingly, our simple linear regression analysis revealed that AM; *Serf2*$^{br-/-}$ mice tend to have a higher plaque load than AM mice with similar A$\beta$ levels (Table S4). A second A$\beta$ staining with the similar W0-2 antibody, directed at amino acid residues 4–10, displayed a similar slight increase in the amount of plaque deposits in AM;*Serf2*$^{br-/-}$ at 3 mo (two-way ANOVA: all 3m non-AM versus AM groups $p_{bon}$ < 0.0001; AM versus AM;*Serf2*$^{br-/-}$ $p_{bon}$ = 0.101) (Figs 5G, S4, and S5B). In fact, a comparison between 6E10 and W0-2 staining in individual animals confirmed the similarity between both staining patterns at 3 mo (Fig S5C and Table S4). Interestingly, our detection algorithm also picked up a small increase in the counts of W0-2-positive foci in the 1 mo AM; *Serf2*$^{br-/-}$ group (Two-way ANOVA: 1m WT versus AM;*Serf2*$^{br-/-}$ $p_{bon}$ = 0.088; *Serf2*$^{br-/-}$ versus AM;*Serf2*$^{br-/-}$ $p_{bon}$ = 0.081; AM versus AM;*Serf2*$^{br-/-}$ $p_{bon}$ = 0.127; all other comparisons $p_{bon}$ > 0.999) (Fig 5G, left panel). Closer examination of the microscopic images revealed the identified spots were not extracellular plaques, but were the result of increased levels of W0-2-positive intracellular A$\beta$ staining in AM;*Serf2*$^{br-/-}$ (Fig S6A). Some low-level intracellular W0-2 and 6E10 staining could also be observed in the other 1-mo-old AM mice, but not to the extent that it was picked up by the detection algorithm (Fig S6A and B). This would suggest that the deletion of *Serf2* already alters the intracellular accumulation of A$\beta$ before plaque deposition in a way that affects antibody binding affinity.

To investigate if SERF2 also affected the amount of Thioflavin-positive fibrils, we performed Thioflavin-S (ThS) staining. Here, we again observed a large variety in the amount of ThS-positive plaques between individual animals, but we found no mean difference between AM and AM;*Serf2*$^{br-/-}$ mice (two-way ANOVA: all 3m non-AM versus AM groups $p_{bon}$ < 0.001) (Fig 5H, left panel). However, further analysis of the correlation between A$\beta$ levels and the amount of ThS-positive plaques revealed that AM mice showed a positive correlation similar to the general A$\beta$ staining, whereas AM; *Serf2*$^{br-/-}$ mice displayed an inverse relationship (Fig 5H, right panel and Table S4). Consequently, a general increase in A$\beta$ deposits does not equate to an increase in ThS-positive amyloid deposition in AM; *Serf2*$^{br-/-}$ mice, as it does in AM mice (Fig S5D). Combining the

unaltered levels of amyloid $\beta$ with changes in the numbers of deposits and their affinity for ThS suggests a change in the amyloid aggregation in *Serf2* KO mice.

## Knockout of Serf2 changes the plaque composition and fibril structure

Given the differences in the antibody and ThS binding affinity of the A$\beta$ deposits, we decided to explore the composition within the individual plaques in greater detail. To this end, we made use of two luminescent conjugated oligothiophenes (LCO), small hydrophobic ligands that bind distinct amyloid structures and can easily be distinguished from each other based on their individual spectral properties (Klingstedt et al, 2011). The smallest, qFTAA, has been shown to stain compact, multifilament fibril structures, coinciding with ThT and Congo red staining (Klingstedt et al, 2011; Nyström et al, 2013; Psonka-Antonczyk et al, 2016). The larger hFTAA is capable of staining prefibrillar (non-thioflavinophilic) and single fibril aggregates, also stained with 6E10, but not Congo red, ThT or qFTAA (Klingstedt et al, 2011; Nyström et al, 2013; Psonka-Antonczyk et al, 2016) (Fig 6A). Previous studies have shown that by combining LCO dyes, it is possible to uncover structural differences in plaque composition, which were shown to be associated with different disease phenotypes in patients (Lord et al, 2011; Nyström et al, 2013; Rasmussen et al, 2017). An exploratory frequency analysis of the 502/588 nm intensity ratios plaques in 3-mo-old AM and AM; *Serf2*$^{br-/-}$ mice revealed a three-peak pattern in which the middle peak was similarly represented in both groups (Fig 6B, green area). The first peak, however, was predominantly found in AM mice (Fig 6B, red area), whereas the third peak was mainly detected in AM; *Serf2*$^{br-/-}$ (Fig 6B, blue area). This shift in ratio was also reflected by a difference in the average intensity ratio between AM and AM; *Serf2*$^{br-/-}$ (*t* test P < 0.001) (Fig 6C). This increase in intensity ratio could not be observed in AM;*Cre*$^{br}$ mice (*t* test P = 0.6047) (Fig 6C, insert), indicating that this effect is caused by the *Serf2* KO and unrelated to *Cre* expression. Given the fact that AM;*Cre*$^{br}$ animals also show a decrease in brain weight (~8% reduction in AM;*Cre*$^{br}$ versus ~11% in the AM;*Serf2*$^{br-/-}$) but do not show similar change in LCO staining, this suggest that the structural changes are also unrelated to the reduction in brain weight. Next, we examined if the change in the intensity ratio correlated with the slight increase in 6E10-positive deposits we observed in AM;*Serf2*$^{br-/-}$ (Fig 6D). However, the higher average intensity ratio in AM;*Serf2*$^{br-/-}$ mice was not related to this general plaque load nor the number of W0-2- or ThS-positive deposits (Fig S5E and F). Further microscopic analysis of low-intensity ratio deposits from AM mice showed a plaque with a qFTAA- and hFTAA-positive core and a border that was only stained by hFTAA (Fig 6E, left panels). The high-intensity ratio deposits from AM;*Serf2*$^{br-/-}$ mice, on the other hand, revealed

**(D)** Correlation plot depicting the relationship between 6E10 plaque density in the ROI and the LCO average intensity ratio in AM (black) and AM;*Serf2*$^{br-/-}$ (grey) mice at 3 mo of age (ROI = cortical and hippocampal area, detailed statistics in Table S4). **(E)** Spectral confocal microscopy images of amyloid plaques in AM and AM;*Serf2*$^{br-/-}$ mice at 3 mo of age double stained with qFTAA and hFTAA (top = fluorescence at 502 nm, middle = fluorescence at 588 nm, bottom = merged image of 502 and 588 nm fluorescence, scale bar = 5 $\mu$m). **(F)** Scanning transmission electron microscopy pictures of amyloid plaques in the cortex of AM and AM;*Serf2*$^{br-/-}$ mice at 3 mo of age (top scale bar 5 $\mu$m, rectangle = zoomed region in bottom picture, bottom scale bar = 1 $\mu$m). High-resolution EM images are available via http://www.nanotomy.org. Source data are available for this figure.

a qFTAA-positive core, but virtually no hFTAA-staining (Fig 6E, right panels), demonstrating the difference in plaque composition reflected by the shift in LCO intensity ratios. In addition, we visualized the global fibrillar structure of the plaques on a nanometer scale using high-resolution scanning transmission electron microscopy (STEM). Overall, STEM images of plaques from AM;*Serf2*$^{br-/-}$ tended to show more condensed plaques composed of short, thick, and densely packed bundles of fibers with little space in between (Fig 6F, right panels, and source data of Fig 6). In the AM mice, however, the plaques we observed displayed more loosely packed fibrils (Fig 6F, left panels). Taken together, these findings suggest *Serf2* KO in mice leads to a shift in the A*β* aggregation process, resulting in an altered structural composition of the plaques.

# Discussion

Previous studies identified MOAG-4 and its human orthologs SERF1A and SERF2 as modifiers of aggregation of amyloidogenic proteins. In this study, we have demonstrated in cells and mice that SERF2 provides a growth advantage during development. The absence of SERF2 or MOAG-4 in itself does not appear to be lethal in cells, worms or the brain-specific KO mice. Therefore, we postulate that the observed perinatal lethality in the full-body KO mice is a secondary effect of the delay in growth due to insufficient maturation of certain organs at birth, rather than indicating an essential gene function. This appears to be supported by the fact that some pups survived the initial minutes after birth and incomplete penetrance of the lethality in the earliest generations with mixed genetic background. The 129SV genetic background present in these mice is known to have a gestation period that is on average around 1 d longer than C57/Bl6J mice (Murray et al, 2010), which could result in improved lung maturation. These findings are also corroborated by another recent study generating SERF2 KO mice (Cleverley et al, 2021).

The effects of SERF2 on growth and development can be caused by several mechanisms. These mechanisms include the facilitation by SERF2 of cell proliferation directly or it protecting cells from stressors that create an unfavorable environment for cell division, both of which would slow general growth in case of SERF2 depletion. In the latter case, we would expect to see an up-regulation of stress pathways or cell cycle arrest proteins inhibiting the cell cycle in *Serf2*$^{-/-}$ cells. Interestingly, the RNA-seq analysis of the MEFs from our *Serf2*$^{-/-}$ mice showed the exact opposite. Cell cycle-driving factors were up-regulated (e.g., cyclins), whereas stress signaling and cell cycle inhibiting factors were down-regulated (e.g., GADD45 and GAS1). Thus, the *Serf2*$^{-/-}$ cells actually displayed an increased stimulation of cell proliferation mechanisms which would fit with cells from earlier stages of embryonic development. Although the exact endogenous biological function of SERF2 remains unclear, a recent study showed that SERF1a might play a role as an RNA-organizing protein, which localizes to membraneless nuclear organelles, like the nucleolus, under normal physiological conditions. However, under stress conditions, it was shown to migrate to the cytoplasm, where it could drive amyloid toxicity (Meyer et al, 2019 *Preprint*). Although there was no compensatory

up-regulation of SERF1 in response to SERF2 KO and a similar mode of action remains to be demonstrated for SERF2, the region responsible for the RNA-binding properties of SERF1A is highly conserved in MOAG-4 and SERF2 (Meyer et al, 2019 *Preprint*). Given the importance of nucleolar disassembly and reassembly in cell cycle control (Visintin & Amon, 2000; Leung et al, 2004), a putative RNA-organizing function could explain how SERF2 facilitates cell proliferation, but its precise function in cell proliferation remains to be identified. Loss of SERF2 function in development and cell growth, in addition, would have limited impact in non-proliferating, differentiated cells, like the brain's neurons or the cells of adult *C. elegans*, and may explain why we saw no adverse effects in the *moag*-4 deletion worms or our adult *Serf2*$^{br-/-}$ mice.

In addition, we confirmed the ability of SERF2 to modify the A*β* pathology in vivo in a mouse model without it changing the overall A*β* levels. This finding is in accordance with the previously observed effects of MOAG-4 on the aggregation of polyglutamine, A*β*, and *α*Syn in *C. elegans* (van Ham et al, 2010). At 1 mo of age, before plaque deposition, we already observe a change in intracellular A*β* accumulation, which resembles the recently reported intracellular, perinuclear accumulation believed to be the precursor for the dense plaque core (Lee et al, 2022). We further showed that mice lacking SERF2 were more prone to form A*β* deposits and that the composition of these deposits was structurally different. In AM mice, higher levels of A*β* result in an increase in the numbers of A*β* deposits and an increase in the number of ThS-positive deposits. AM;*Serf2*$^{br-/-}$ mice, on the other hand, also show an increase in A*β* deposits, but this did not lead to higher numbers of ThS-positive plaques, indicating an altered dynamic and outcome of amyloid formation. These findings were further corroborated by the LCO spectra, which revealed that the plaques in AM;*Serf2*$^{br-/-}$ mice have a different conformation of pre-fibrillar and fibrillar A*β* compared with AM mice. Finally, STEM imaging also confirmed a globally altered structure of the amyloid fibrils in the plaques.

Based on previous studies, the LCO spectra observed in brains of *Serf2*-deletion mice, however, would resemble the spectra of increased mature fibrils (Nyström et al, 2013; Psonka-Antonczyk et al, 2016). If so, this would seemingly contradict findings in previous studies, such as the in vitro aggregation results in Pras et al (2021). However, with a finite amount of aggregating protein, SERF2 mainly acts on nucleation. It is therefore possible that removal of *Serf2* could shift the balance in the aggregation kinetics towards elongation and maturation instead of the formation of new nuclei, resulting in more mature fibrils. Further in-depth temporal and structural analyses, including those assessing associations between SERF2 and amyloid *β* in brain, will be required to determine mechanism by which depletion of *Serf2* altered the amyloid structures.

Amyloid fibrils are formed through a nucleated self-assembly process characterized by a structural conformation to a *β*-strand secondary structure and the formation of a critical nucleus during the initial oligomerization. The nuclei act as seeds from which the fibrils grow and have been shown to propagate a specific fibril structure (Petkova et al, 2005; Qiang et al, 2017). In vitro kinetic assays SERF and MOAG-4 accelerate aggregation by acting on the nucleation phase (Falsone et al, 2012; Yoshimura et al, 2017; Meinen

et al, 2019; Merle et al, 2019). Recently, we have shown that SERF2 is equally capable to drive amyloid formation of α-Syn and Aβ in vitro through a conserved, positively charged region (Pras et al, 2021). Meanwhile, another study has demonstrated that binding of the intrinsically disordered protein yeast SERF to α-Syn and Aβ results in fuzzy complexes with heterogeneous structural conformations that are more extended in nature (Meinen et al, 2019). Intra-molecular electrostatic interactions have also been proven to play a part in the dynamics and structure of the Aβ monomer folding, which is at the basis of nucleus formation. In fact, several of the familial AD mutations located within the Aβ sequence and reported not to affect the Aβ levels appear to exert their effects by modifying the intramolecular interactions and monomer folding (Grant et al, 2007; Lazo et al, 2009; Ni et al, 2011; Elkins et al, 2016). These mutations have been shown to alter the aggregation kinetics, fibril structure, and localization of Aβ accumulation and deposition, giving rise to specific disease phenotypes in mice and human patients (Miravalle et al, 2000; Grabowski et al, 2001; Nilsberth et al, 2001; Van Nostrand et al, 2001; Lord et al, 2006, 2011; Grant et al, 2007; Philipson et al, 2009; Bugiani et al, 2010; Tomiyama et al, 2010; Inayathullah & Teplow, 2011; Ni et al, 2011; Ovchinnikova et al, 2011; Shimada et al, 2011; Kulic et al, 2012). The differences in Aβ pathology between these mutants seem to display some similarities to the changes we observed between our AM and AM;Serf2$^{br-/-}$ mice. Together with the in vitro findings about the mechanism of SERF2 interaction with aggregation-prone proteins, this would appear to suggest the modulation of intramolecular interactions and altered nucleation as the mechanism for SERF2's effect on aggregation and amyloid formation in our mice.

Interestingly, there is increasing evidence that qualitative, structural properties may be more related to toxicity than the quantity of aggregates and deposits. Recent studies have provided increasing evidence that structural polymorphs are associated with differences in toxicity and different clinical phenotypes in sporadic and familial cases (Cohen et al, 2009, 2015; Lu et al, 2013; Qiang et al, 2017; Rasmussen et al, 2017). One recent study even demonstrated that these structural conformations of distinct disease phenotypes could also be detected by differences in the LCO spectra of the plaques and that these spectral properties could, at least partially, be transmitted to a mouse model through seeding (Rasmussen et al, 2017).

Although the effect of familial mutations on the formation of distinct polymorphs has already been explored by others (Lord et al, 2006; Kulic et al, 2012; Philipson et al, 2012; Rasmussen et al, 2017), our study provides the first evidence of a single endogenous factor, separate from Aβ and its production pathway, contributing to a structural shift of amyloid pathology in a mammalian system. Further research will be needed to elucidate the exact structural changes at an atomic level and if they affect toxicity and disease progression similarly to what was previously observed in *C. elegans*. This will provide new insights into the structural properties and diversity of disease–protein aggregation, contributing to a better understanding of the variability in disease manifestation and opening up previously un-explored avenues for therapeutic research. In the case of SERF2 specifically, its apparent antagonistic pleiotropy and potentially reduced biological importance in later life could prove interesting with regard to the treatment of age-related neurodegenerative disorders. However, further exploration of SERF2's endogenous function and how it evolves with ageing will be needed to fully assess this mechanism's therapeutic potential.

# Materials and Methods

### Animals

*Serf2* knockout mice were generated by introducing *loxP* sites in the *Serf2* gene on either side of exon 2, in accordance with the "knockout-first" allele design as described by Skarnes and colleagues (Skarnes et al, 2011). Full-body *Serf2* knockout mice were obtained by crossing these floxed mice with 129SV mice expressing *Cre* under the *Hprt* promoter, resulting in the removal of exon 2 by *Cre*-mediated recombination (Fig S1A). Subsequently, these mice with a mixed background were backcrossed at least six times to a C57Bl/6J background. Homozygous and heterozygous full-body *Serf2* knockout mice were examined with their wild-type littermates at embryonic days 13.5, 15.5, and 17.5, and at the day of birth (P0) and day of weaning (P22).

Heterozygous *Sox1-Cre* mice (Takashima et al, 2007) were backcrossed at least eight times to a C57BL/6J background. These animals were also used as controls to differentiate between effects of *Cre* expression and Serf2 knockout. To generate the conditional *Serf2* knockout mice, the floxed mice were first crossed with Tg (ACTFLPe)9205Dym (#003800; Jackson) mice to flip the FRT site and remove the lacZ and neo cassette. Subsequently, these mice were backcrossed at least 6 times to a C57BL/6J background. Finally, the resulting homozygous *Serf2*$^{flox/flox}$ mice were crossed with the backcrossed *Sox1-Cre* mice to obtain the brain-specific *Serf2* knockout mice (Fig S3A). The conditional *Serf2* knockout mice were crossed with the APPPS1-21 transgenic mice (APP (KM670/671NL)/ PS1(L166P)) (Radde et al, 2006). A final cross between heterozygous APPPS1-21; *Serf2*$^{flox/flox}$, and heterozygous *Sox1-Cre*; *Serf2*$^{flox/flox}$ resulted in four experimental groups of *Serf2*$^{flox/flox}$ mice: *Sox1-Cre*$^{-/-}$; APPPS1-21$^{-/-}$ (WT), *Sox1-Cre*$^{+/-}$; APPPS1-21$^{-/-}$ (*Serf2*$^{br-/-}$), *Sox1-Cre*$^{-/-}$; APPPS1-21$^{+/-}$ (AM), and *Sox1-Cre*$^{+/-}$; APPPS1-21$^{+/-}$ (AM;*Serf2*$^{br-/-}$) (Fig 2A). All experimental groups were tested at 1 or 3 mo of age. *Serf2*$^{br-/-}$ for behavioral phenotyping resulted from breeding conditional *Serf2* knockout mice with APPPS1 transgenic mice (APP(KM670/671NL)/ PS1dE9) (Borchelt et al, 1996) and phenotyping started at the age of 9 wk.

All experiments were approved by the Institutional Animal Care and Use Committee of the University of Groningen (Groningen, The Netherlands) and by the responsible authority of the district government of Upper Bavaria, Germany.

All mice were maintained on a C57BL/6J background and housed in a 12:12 h light/dark cycle and the animals had ad libitum access to food and water. Genotyping of embryos was performed using tail and yolk sac biopsies. Otherwise, ear biopsies from labelling were used. DNA was purified using prepGEM Tissue kit according to a protocol adapted from the manufacturer (ZYGEPTI0500, ZyGEM; VWR International BV) and subjected to PCR using the primers listed in Table S5. If not otherwise stated, the mice were terminated

through $CO_2$ inhalation and cervical dislocation for subsequent tissue collection.

### Embryo processing and histochemistry

Embryos were fixed in 4% formalin (Kinipath) for a minimum of 24 h at room temperature. For the pathological analysis, the embryos and their placenta were bisected longitudinally and embedded in paraffin. Using the microm HM 340E (Thermo Fisher Scientific), 4 $\mu$m sections were cut for the hematoxylin–eosin (HE) staining. The HE sections were incubated at 60°C for 15 min. Next, sections were deparaffinized and rehydrated in xylene (2×), 100% alcohol (2×), 96% alcohol, 70% alcohol, and water. HE staining was performed by incubation with hematoxylin for 4 min, water for 10 min, eosin for 1 min, and water for 10 s. After staining, all sections were dehydrated in 70% alcohol, 96% alcohol, 100% alcohol (2×), and xylene (2×).

### Brain processing

Upon termination, all brains were collected and weighed. From each experimental group, three whole brains were collected for HE staining according to the protocol described under embryo processing. HE-stained sections were scanned with the TissueFAXs microscope using 20X objective. Images were processed for nuclei counts using Histoquest software. The remaining brains were divided sagittally. The left hemibrain was prepared for histochemistry and fixed in 4% PFA for 48 h at room temperature on a shaker. Next, the brains were placed in 30% sucrose for ~12 h. Afterwards, the excess sucrose was removed and the hemibrains were frozen in molds with Tissue Tek O.C.T. compound (Sakura) on dry ice. The right hemibrain was snap frozen in liquid nitrogen and stored at –80°C. This tissue was homogenized using a liquid nitrogen-cooled mortar for subsequent protein and RNA analysis.

### Quantitative RT–PCR

Total RNA was extracted from snap frozen tissue using TRIzol Reagent (Life Technologies) according to the manufacturers' description. Total RNA quality and concentration were assessed using a NanoDrop 2000 Spectrophotometer (Thermo Fisher Scientific/Isogen Life Science). cDNA was made from 1.5 $\mu$g total RNA with a RevertAid H Minus First Strand cDNA Synthesis kit (Thermo Fisher Scientific) using random hexamer primers. Quantitative real-time PCR was performed using a Roche LightCycler 480 Instrument II (Roche Diagnostics) with SYBR green dye (Bio-Rad Laboratories) to detect DNA amplification. Relative transcript levels were quantitated using a standard curve of pooled cDNA solutions. Expression levels were normalized to $\beta$-Actin or 18S mRNA levels. The primers for quantitative PCR used are listed in Table S5.

### Western blot analysis

For SERF2 analysis tissues were homogenized in RIPA buffer (50 mM Tris pH 8, 150 mM NaCl, 5 mM EDTA, 0.5% SDS, 0.5% SDO, 1% NP-40) with protease inhibitor cocktail (Roche) and incubated on ice for 1 h,

spun down at 17,000$g$ for 30 min at 4°C, and the supernatant was collected. Protein measurements were performed using a BCA kit (Pierce) and 150 $\mu$g was loaded on a 10–20% tris/tricine SDS–PAGE gels (Bio-Rad Laboratories) and blotted onto 0.2 $\mu$m nitrocellulose membrane (Bio-Rad Laboratories). Membranes were incubated overnight with SERF2 (1/1,500; Protein tech) or actin (1/10,000; MP biomedicals) antibody. Next, the membranes were incubated with anti-mouse or -rabbit secondary antibodies tagged with horseradish peroxidase (1/10,000; Bio-Rad Laboratories) for 1 h at room temperature and visualized by chemiluminescence (Amersham ECL prime Western blotting detection reagent; VWR).

### RNA sequencing

RNA sequencing analysis was performed on three MEF cell lines from $Serf2^{-/-}$ mice and four $Serf2^{+/+}$ littermate controls. Total RNA was isolated from MEFs using the QIAGEN RNeasy isolation kit. Integrity of the RNA was based on RIN scores as determined by a Bioanalyzer (Agilent). RNA-sequencing libraries were prepared using TruSeq Stranded Total RNA with Ribo-Zero Human/Mouse/Rat (RS-122-2201; Illumina) according to manufacturer's protocol. Pooled libraries were sequenced on an Illumina HiSeq 2500 (single-end 50 bp). Reads were aligned to the mouse reference genome (mm10) using a splicing-aware aligner (StarAligner). Aligned reads were normalized to fragments per million (FPM), excluding low abundance genes (mean FPM > 1 in at least two samples). The raw count data were preprocessed using the programming language R (3.4.0) (R Core Team, 2016, available online at: www.r-project.org), the program RStudio (1.0.143) (RStudio Team, 2016, available online at: http://www.rstudio.com/), and the EdgeR package (3.18.0) (Robinson et al, 2010). Genes that displayed FPM value > 1 in at least two libraries were retained, resulting in a list of 12,808 genes for differential analysis. Differentially expressed (DE) genes between the $Serf2^{+/+}$ and $Serf2^{-/-}$ MEFs were identified using the EdgeR general linear model approach. After statistical correction for multiple comparisons with the "false discovery rate" (FDR) method (FDR < 0.05), a list of 738 DE genes was obtained. DAVID (6.8) was used to perform functional annotation analyses on this gene list and to identify significantly enriched gene ontology (GO) terms (using GOTERM_BP_DIRECT) (Huang et al, 2009a, 2009b). The connectivity between the enriched GO terms was further examined by determining the amount of associated genes found (AGF) that were shared between two GO terms. The most significant relationships (where a GO term shared > 25% of the AGF) were mapped in a network using the igraph package (1.0.1). The color of the edges reflects the major node, with the largest amount of AGF in each relationship. The weight of the edges was determined by the percentage of AGF shared by the minor node in the relationship. We defined three clusters of GO terms, where multiple minor GO terms connected to one or two central, major GO terms. Minor GO terms always clustered with the major GO term with which they shared the strongest connection. One GO term, positive regulation of the apoptotic process, showed no direct relation with one of the major GO terms and was therefore assigned to the only cluster it had a significant connection with (Fig S2A and B).

## Cell culture

MEFs were isolated from $Serf2^{+/+}$, $Serf2^{+/-}$, and $Serf2^{-/-}$ E13.5 embryos. MEFs were cultured in T75 culture flasks (658175; Greiner Bio-One), high-glucose DMEM (Gibco), supplemented with 10% FBS (12133C; Sigma-Aldrich), 1% penicillin/streptomycin (Gibco), non-essential amino acids (Gibco) and $\beta$-mercaptoethanol at 37°C, 5% $CO_2$, and 3% $O_2$. E13.5 embryos. Wild-type HEK293T and two independent clonal lines of CRISPR $Serf2$ KO mutant HEK 293T cells were cultured in DMEM (high glucose, pyruvate, 41966052; Gibco), supplemented with 10% Bovine Cow Serum (BCS; 12133C; Sigma-Aldrich) and 1% penicillin/streptomycin (10,000 U/ml, 15140122; Gibco), at 37°C, 5% $CO_2$. For passaging of cells, 0.05% Trypsin–EDTA (1X), Phenol red (25300-054; Invitrogen) was used. Regular mycoplasma tests were performed.

## Cell proliferation assay

For the proliferation assay, 200,000 cells of each cell line were plated in triplicate in a standard 12-well plate and incubated inside the IncuCyte Zoom Live-Cell Imaging System (Essen Bioscience) for over 100 h. The Incucyte system captured phase-contrast images with a 10X magnification every 2 h at nine distinct locations in each well to determine average confluency as a measure of cell growth. The quantification was performed using the IncuCyte Zoom software. The growth experiment was repeated three times. The confluency data of each experiment were normalized by min–max scaling and the average of the three experiments was plotted (solid line) with the SD (dashed lines).

## Mouse phenotyping pipeline

30 Serf $^{br-/-}$ (15 male and 15 female mice) and 30 littermate controls (male and female mice) underwent to a neuro-behavioral phenotypic screening pipeline at the German Mouse Clinic (Gailus-Durner et al, 2005; Fuchs et al, 2018) (see also www.mouseclinic.de). Measurements were taken from weeks 9–29 wk. Experimental groups were assigned according to the genotype of the animals. Metadata for each data point were recorded throughout the measurements. The phenotypic tests were performed according to the standardized protocol as described before (Fuchs et al, 2011; Garrett et al, 2012; Hölter et al, 2015; Salminen et al, 2017; André et al, 2018; Heermann et al, 2019) and available as extended material & methods. Animal numbers may vary depending on the test performed.

## ELISA

The ELISA experiments were performed as described previously (Huang et al, 2015). Briefly, the frozen brain samples were homogenized in a tissue protein extraction reagent (Pierce) supplemented with complete protease inhibitors (Roche) and centrifuged for 1 h at 430,000$g$ at 4°C. The supernatant was used for ELISA. The $A\beta_{40}$ and $A\beta_{42}$ levels were determined by standard sandwich ELISAs using end-specific antibodies (Janssen Pharmaceutical), and the monoclonal JRFcA$\beta_{40}$/28 and JRFcA$\beta_{42}$/26 antibodies as the capture antibodies. Horseradish peroxidase-conjugated JRFA$\beta$N/25 or JRFA$\beta$N/25 were used as the detection antibodies for, respectively, human A$\beta$ or murine A$\beta$. Samples were loaded in triplicate for each ELISA. The ELISAs were performed in triplicate for the 3-mo-old animals and in duplicate for the 1-mo-old animals.

## Brain immunohistochemistry

To evaluate A$\beta$-related pathology, the brain was cut in 30 $\mu$m-thick serial cryostat sections. A series of sections was selected with 300 $\mu$m spacing between individual sections and blind-coded, six per mouse for the 6E10 and Iba1 antibody and three per mouse for W02. Sections were fixed in 4% PFA for 10 min, followed by 20 min antigen retrieval in citrate acid (pH 6.0) and 30 min in 1% $H_2O_2$ in methanol. Next, the sections were incubated for 1 h in 10% goat or donkey serum, depending on the secondary antibody, and immunolabeled overnight with antibodies against A$\beta$ and Iba1 (Table S6). The sections were washed and incubated with the complementary biotinylated secondary antibody (1/500; Jackson Immunoresearch) for 2 h. Next, sections were incubated in ABC complex and reacted with Diaminobenzidine (0.5 mg/ml H2O with 0.01% $H_2O_2$). The stained sections were scanned with the TissueFAXs microscope using 20X objective lens. All images were analyzed using the Fiji platform of the ImageJ software (Schindelin et al, 2012). For unbiased feature detection, we used a single automated script with an optimized threshold method and parameters for the complete image set of each staining method.

## Thioflavin-S staining

A 300-$\mu$m spaced series of three blind-coded 30 $\mu$m sections were fixed in 4% PFA for 10 min, followed by incubation in 0.25% potassium permanganate solution for 15 min, and a bleaching step with 1% potassium metabisulfite/1% oxalic acid for 5 min, incubation with 0.02% Thioflavin-S solution in 50% ethanol (T1892; Sigma-Aldrich) for 8 min, rinsed with water between every step. Finally the sections were incubated with 1 $\mu$l/ml DAPI. The sections were analyzed as described for the immunostained sections. Stained sections were scanned with the TissueFAXs microscope using 20X objective lens. Images were processed using Tissuequest software, selecting the cortex for automated analyses.

## LCO staining

For the LCO stainings, two different LCO variants, qFTAA (quadro-formyl thiophene acetic acid) and hFTAA (hepta-formyl thiophene acetic acid) were used. Blind-coded 30 $\mu$m sections were double-stained with qFTAA and hFTAA (2.4 $\mu$M qFTAA and 0.77 $\mu$M hFTAA in PBS) similar to a previous description (Nyström et al, 2013; Rasmussen et al, 2017). Sections were incubated for 30 min in the dark at room temperature. The stainings were analyzed on the Zeiss LSM 510 META confocal microscope equipped with an argon 458 nm laser (Carl Zeiss MicroImaging GmbH). A 40× objective lens (oil-immersion, 1.3 NA; Zeiss) was used for spectral imaging of the A$\beta$ deposits. Stacked images were acquired every 10 nm in the emission spectrum between 470 to 695 nm. Locations were selected randomly from the plaque containing regions of the temporal, frontal, and occipital cortices. Images were analyzed using Fiji

(Schindelin et al, 2012, 2015) and standard computer algorithms to provide a reproducible, unbiased detection of the plaques. First, the stacks of images across the different wavelengths of the spectrum were combined into one image using the Z-projection sum. A threshold was applied to the resulting image using the "Triangle dark" algorithm to automatically identify and delineate the plaques. Partial plaques on the edge of the images and particles smaller than 25 $\mu m^2$ were excluded. These settings allowed us to detect over 98% of all the plaques in the images. The missed plaques were either too small or too low in intensity to be accurately distinguished from the background. Incorrectly identified particles because of (lipofuscin) auto fluorescence were readily identified based on their wavelength intensity profile and visual inspection and were also excluded from the analysis. We identified between 17–25 plaques in each animal for the spectral intensity measurements. The ratio of the intensity of emitted light at the blue-shifted portion (502 nm) and red-shifted peak (588 nm) was used as a parameter for spectral distinction of different A$\beta$ deposits. These peaks of the spectra were selected to maximize the spectral distinction.

### Large-scale STEM (nanotomy)

The STEM experiments were adapted from previously described experiments (Kuipers et al, 2015). Briefly, paraffin embedded cortical sections of AM and AM;$Serf2^{br-/-}$ were deparaffinized and postfixed with 1% osmium tetroxide/1.5% potassium ferrocyanide in 0.1 M sodium cacodylate, dehydrated through ethanol, and embedded in EPON (Serva) using a tissue processor (EM TP 709202; Leica). Ultrathin sections (80 nm) were cut using the Leica uc7 ultramicrotome and collected on formvar-coated cupper grids (electron microscopy sciences). A large area scan using scanning transmission detection was made using a Zeiss supra55 SEM with ATLAS. STEM detection with a four-quadrant STEM detector was used in inverted dark-field mode, at 28 kV with 30 $\mu m$ aperture at 3.5 mm working distance. All images were recorded at the same scan speed (cycle time 1.5 min at 3,072 × 2,304 pixels). Contrast and brightness were set based on a live histogram. High-resolution large-scale STEM images at 2.5 nm pixel size were generated with the external scan generator ATLAS (Fibics), individual tiles were stitched in VE viewer (Fibics), exported as a html file, and uploaded to the website www.nanotomy.org.

### Statistical analysis

Unless specified otherwise in an experiment-dedicated M&M section, like in the RNA sequencing section, statistical analysis was performed using Graphpad v7.02. All correlation analyses were performed using the simple linear regression. The exact statistical test used for the other comparison is specified in the main body of the text, together with the resulting $P$-values. In the case of multiple comparisons between genotypes, a post-hoc analysis with Bonferroni correction was performed. Any corrected $P$-values derived from this post-hoc analysis are indicated in the text as $p_{bon}$. Exact $P$-values are provided except for extreme $P$-values

exceeding $P < 0.0001$ or $P > 0.9999$. An $\alpha$ value of 0.05 was maintained for both hypothesis and multiple correction testing.

## Data Availability

RNA-seq data have been deposited in the ArrayExpress database at EMBL-EBI (www.ebi.ac.uk/arrayexpress) under accession number E-MTAB-10083. Nanotomy datasets are open access available via the repository website www.nanotomy.org.

## Supplementary Information

## Acknowledgements

All LCOs were kindly provided by K Peter R Nilsson, Linköping University, Sweden. Part of the work has been performed in the UMCG Microscopy and Imaging Center (UMIC), sponsored by ZonMW grant 91111.006, NWO 175-010-2009-023, STW 12718, NWO 91116005, NWO 40-00506-98-9021 (NEMI), NWO National Roadmap for Large-Scale Research Infrastructure of the Dutch Research Council (NWO 184.034.014). We thank Ben Giepmans for advice on EM studies. This project was funded by a Meervoud Grant from NWO (836.09.001) (to EAA Nollen), a European Research Council (ERC) starting grant (281622 PDControl) (to EAA Nollen), the Alumni chapter Gooische Groningers facilitated by the Ubbo Emmius Fonds (to EAA Nollen), an Aspasia fellowship from NWO (015.014.005) (to EAA Nollen), a CORBEL PID 2311 grant (to EAA Nollen) and the German Federal Ministry of Education and Research (Infrafrontier grant 01KX1012) (to M Hrabe de Angelis).

### Author Contributions

E Stroo: conceptualization, validation, investigation, visualization, methodology, and writing—original draft.
L Janssen: conceptualization, validation, investigation, visualization, methodology, and writing—original draft.
O Sin: conceptualization, formal analysis, investigation, and writing—review and editing.
W Hogewerf: conceptualization, data curation, formal analysis, investigation, methodology, and project administration.
M Koster: validation, investigation, visualization, and methodology.
L Harkema: formal analysis and investigation.
SA Youssef: formal analysis, investigation, and methodology.
N Beschorner: resources, formal analysis, investigation, visualization, and methodology.
AHG Wolters: data curation, formal analysis, investigation, and visualization.
B Bakker: data curation and formal analysis.
L Becker: data curation, investigation, and methodology.
L Garrett: data curation, formal analysis, and methodology.
S Marschall: conceptualization, resources, data curation, investigation, methodology, and writing—review and editing.
SM Hoelter: conceptualization, funding acquisition, and project administration.

W Wurst: conceptualization and project administration.

H Fuchs: conceptualization and project administration.

V Gailus-Durner: conceptualization, resources, data curation, supervision, funding acquisition, validation, methodology, project administration, and writing—review and editing.

M Hrabe de Angelis: conceptualization, investigation, methodology, and project administration.

A Thathiah: investigation, visualization, and methodology.

F Foijer: conceptualization and supervision.

B van de Sluis: conceptualization and resources.

J van Deursen: resources.

M Jucker: conceptualization, supervision, funding acquisition, and project administration.

A de Bruin: conceptualization, resources, supervision, methodology, project administration, and writing—review and editing.

EAA Nollen: conceptualization, resources, data curation, supervision, funding acquisition, and writing—original draft, review, and editing.

## Conflict of Interest Statement

The authors declare that they have no conflict of interest.

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
