## [Reviewer comments · Life Science Alliance]

Life Science Alliance

Deletion of SERF2 in mice delays embryonic development and alters amyloid deposit structure in brain

Esther Stroo, Leen Janssen, Olga Sin, Wytse Hogewerf, Mirjam Koster, L Harkema, S.A. Youssef, Natalie beschorner, Anouk Wolters, Bjorn Bakker, Lore Becker, Lilian Garrett, Susan Marschall, Sabine Hölter, Wolfgang Wurst, Helmut Fuchs, Valerie Gailus-Durner, Martin Hrabe de Angelis, Amantha Thathiah, Floris Foijer, Bart van de Sluis, Jan van Deursen, Matthias Jucker, Alain De Bruin, and Ellen Nollen

DOI: <https://doi.org/10.26508/lsa.202201730>

Corresponding author(s): *Ellen Nollen, University Medical Center Groningen and Leen Janssen*

Review Timeline:

Submission Date:	2022-09-20
Editorial Decision:	2022-10-28
Revision Received:	2023-03-30
Editorial Decision:	2023-04-11
Revision Received:	2023-04-18
Accepted:	2023-04-18

Transaction Report:

October 28, 2022

Re: Life Science Alliance manuscript #LSA-2022-01730-T

Dr. Ellen AA Nollen
UMC Groningen
European Research Institute for the Biology of Ageing (Eriba)
PO Box 196
Groningen 9700AD
Netherlands

Dear Dr. Nollen,

Thank you for submitting your manuscript entitled "SERF deletion modifies amyloid aggregation in a mouse model of Alzheimer's disease" to Life Science Alliance. The manuscript was assessed by expert reviewers, whose comments are appended to this letter. We invite you to submit a revised manuscript addressing the Reviewer comments.

Thank you for this interesting contribution to Life Science Alliance. We are looking forward to receiving your revised manuscript.

Sincerely,

B. MANUSCRIPT ORGANIZATION AND FORMATTING:

Reviewer #1 (Comments to the Authors (Required)):

This manuscript presents a detailed analysis of SERF2 total and conditional knockout mice, and an analysis of how loss of this protein affects the well-studied APP-PS1 mouse. The characterizations appear to be appropriately done, and a suite of dye staining approaches have been used to quantify the (rather subtle) changes in plaque morphology and fibril structure that occur in the SERF2 conditional knockout background. However, what is most striking is the absence of two seemingly obvious experiments: using immunohistochemistry to assay the distribution of SERF2 in the APP/PS1 background (e.g., is SERF2 associated with intracellular Abeta or extracellular plaques?), and doing behavioral tests to assay cognition (e.g., fear conditioning or water maze tests). There are multiple commercial sources of SERF2 antibodies, and it is hard to understand why the extensive behavioral characterization of the SERF2 KO mice did not include standard measurements of cognition. It is also surprising that no cognitive tests were done on the SERF2 KO/APP-PS1 mice, as this is the standard readout to determine if alterations in the APP- PS1 mice have AD-relevant outcomes. It is unclear to me why a paper with 25 authors did not attempt these experiments, particularly the antibody staining, as this is critical to support their model that SERF2 directly interacts with Abeta to influence its aggregation. (Extrapolating from in vitro experiments is not compelling when they can be tested in vivo.)

Reviewer #2 (Comments to the Authors (Required)):

SERF is a protein originally discovered by the corresponding author of this paper because knockouts in this gene (originally termed MOAG-4) showed a decreased ability to form puncta in a YFP worm model of Huntington's disease. These authors and others have shown that SERF related proteins accelerates amyloid formation in vitro. In an attempt to establish the normal in vivo function of this protein, the authors genetically knocked out serf2 in mammalian cell lines and mouse models, and did a detailed characterization of their phenotypes. In addition to the whole-body knockout, the authors generated a brain-specific serf2 knockout mouse, and looked at phenotypes associated with this serf2 knockout, then examined the impact of SERF2 on amyloid plaque deposition using the Alzheimer's mouse model, APPPS1. The most striking phenotype is that the whole-body mouse knockout of SERF2 is perinatally lethal, due to a lung developmental defect. Brain specific knockouts no major neurological defects, do not impact APP levels (SuppFigG-I) or AB production (Fig5C-E), but rather, showed surprisingly subtle phenotypes serf2, loss of serf2 showed minor behavior defects, namely a heightened level of agitation and hyper-responsiveness and loss of serf2 may have altered the morphology of amyloid plaques. •

Brain-specific loss of serf2 alters the number of AB deposits in mice brains (Fig5F-H, SuppFig3). The authors verified previous indications that SERF2 acts cell autonomously, and found that serf1a levels do not change to compensate loss of serf2 perhaps suggesting non-complementary roles.

Given the history of SERF's discovery and characterization so far, it is understandable that the author's chose to focus on investigating the nerve specific phenotypes of SERF knockout, but they seem to have missed an opportunity to understand the strong perinatal lethality phenotype by only conducting a preliminary RNAseq analysis. Much of the published literature regarding serf focuses on SERF1a, not SERF2 and the author's data sheds some light on what is still unknown regarding the impact of serf proteins on protein aggregation. In the opinion of this reviewer, papers such as this that present overall mostly negative results are worthy of publication because they have the potential of redirecting the field towards more productive directions. However, the rationale of the underlying mechanism(s) to explain the knockout phenotypes are weak and sometimes in apparent partial contradiction to their own previous findings, so further discussion and clarification is in order. This shouldn't be an attempt to shoehorn these recent findings into existing literature but rather an honest effort to advance the understanding of the function(s) of this interesting protein.

Detailed comments:

The title:

"SERF deletion modifies amyloid aggregation in a mouse model of Alzheimer's disease" might be made less declarative because of concerns expressed below.

Line 50 the word "show" is probably too strong here as different binding by structure specific amyloid dyes really just provides a preliminary indication that the structure of amyloid fibrils are different.

Line 51-52 Blinded studies revealing statistically significant differences in fiber morphology would be far more convincing that the two images of fiber plaques shown, which are not convincing at all. What is needed would be either stats based on blind

observers, or a lot of examples in the supplement plus a much more couched statement here. Something like "TEM imaging suggests that the structure of amyloid deposits could be altered by SERF2 knock-out, and further study would be needed to confirm this observation." The authors show just one example. This is definitely not enough to be remotely convincing given how easy it is to bias TEM data.

Line 53 no aging studies were done so the words "and aging" should be deleted.

Line 83-84, these lines are more appropriately nuanced and less declarative than the title, the title should be revised accordingly.

Line 86 That "Full-body Serf2 KO results in developmental delay and perinatal lethality in mice" is the strongest finding of the paper, but is not mentioned in the abstract.

Line 101 This is unclear and needs to be clarified. I don't understand how it is justified to lump two different values and compare them to one, an explicit comparison between +/- and +/+ to back up the 50% claim is necessary, is this decrease statistically significant? it seems like there is the claim in the figure with the 3 stars designation but is this to the wt or to the KO? Adding the lines between all the groups being compared in the graph would make this much clearer. It appears like the authors did what was appropriate, mathwise: ANOVA for all groups, then a posthoc test for individual comparisons.

Their text here...

(One-way ANOVA: Serf2+/- and Serf2-/- vs Serf2+/+)

...would be read/interpreted like this:

One-way ANOVA was used to compare the mean mRNA levels within all genotypes to look for the presence of any differences. A posthoc test revealed statistically significant differences specifically between both the Serf2+/+ vs Serf2 +/- and between Serf2+/+ and Serf2 -/-.

Line 102 is similarly brief and needs to be explicitly clarified

Loss of serf2 (whole body and brain) resulted in a smaller mass of the whole body or the brain, respectively

To show that the loss of serf2 resulted in reduced embryo size, authors included measurements of body length (Fig1E) and body mass (SuppFig1E). While there is a significant difference at E17.5 for the length measurements and not body mass (though body mass becomes significantly different between serf -/- and both serf+/- serf +/+ at P0), the differences in body mass on P0 is more convincing than a difference in body length at E17.5

Line 107 The statistics for E15.5 are missing from the graph, these should be done and shown. If the authors did the ANOVA for E15.5, the simplest thing is to add a bar with "ns" for not significant. Importantly the authors don't ever seem to add an "ns" bar anywhere in any of the figures, and while that does keep the graphs cleaner, it can just as easily be interpreted as a lack of significance testing.

Line 108 The actual p-values should be listed throughout the manuscript not just < values. This is particularly important for interpretation when the p-value is near alpha (0.05 in this case). For the same reason, "not significant/ns" should also be an actual value. That gives the reader the ability to see that 0.051 and 0.049 are actually quite close instead of ns vs *.

Line 109 "embryos during the embryonic stage leading up to birth (P0)" needs to be clarified, I think that what is meant is that the size differences were not apparent at any tested embryonic stage, including P0, for hets and homozygous positive genotypes.

Line 111 cause is misspelt

Line 113 "indicates" is too strong, "suggests" is a better choice

Line 122 the phenotypes if any of serf+/- pups should be disclosed and discussed.

The rationale in Lines 123-134 and Figure 2 is not clear nor convincing to explain the phenotypes they observed in cell line knockout and mouse knockout. With the upregulation of cell division factors and downregulation of growth arrest factors in serf2-/- MEFs shown by RNA-seq, wouldn't the authors expect to see faster cell division in these serf2-/- MEFs compared to WT MEFs? How does this RNA-seq data explain the slower proliferation they observed in serf2-/- HEK293T cells?

Line 125 the words cell intrinsic are misleading, I think what is meant is cell autonomous.

Although they show that Serf2-/- HEK293T knockout cell lines have reduced proliferation (Fig2A) similar results have been shown before, they did reference van Ham 2010, which showed that serf/moag-4 acted in a cell autonomous manner, but authors did not reference Tsuboyama 2020 which also used CRISPR/Cas9 to knockout serf2 in HEK293T cells, and showed a growth defect, this paper needs to be cited.

Line 129 I found this rather superficial analysis unfortunate. One would hope that authors would follow where data leads, overall impression that they were more intent on proving a connection to amyloidogenesis than following the new observations in their data.

Line 137-139 I found this rationale and transition rather strange, it appears that the authors are more intent on accumulating data in favor of their initial hypothesis, that SERF is involved in amyloid formation, rather than following the clearest effects of SERF knockouts which seem to be on development.

Line 154 This is the first time I've ever seen an interaction term from an ANOVA used correctly in paper, nice!

Line 161-165 It is unclear if the cre effect is sufficient to explain the whole effect or not, until this is explicitly shown is not the references to reduced weight should be deleted. Because of the way the authors split up these data (WT vs Cre, then WT vs Serf2) it probably means they did the experiments at different times and cannot compare directly. Alternatively one might be able to compare normalized data (normalized differences from WT at each age for each tested genotype) could be used in a hypothesis test between genotypes to answer that.

Figure 4 I'm a little surprised the authors don't refer to Pbon for these. Or some other multiple-testing-correction.

Line 185-87. The authors seem appropriately cautious in the interpretation of the behavioral data

Line 188 "proceed with this model" What model exactly?

Line 242 the slight increase in plaque amounts is really slight. Yes, Pbon is over 0.05, but, it is quite common to hold the multiple-testing-correction p's to a different standard than the initial p-values (for too many reasons for this to list here). A way to

deal with this in a paper is to say our alpha for hypothesis testing was 0.05, and our alpha after multiple-testing-correction was 0.1. (Which is routine in bioinf papers)

Line 247 Is this small increase statistically significant? See prev comment. "Statistically (in)significant" is a prior agreement with the reader on an arbitrary alpha value. The standard is 0.05 for a hypothesis test, and but is not standardized for multiple-testing-corrections. They need to spell out precisely what they're threshold is.

Line 286 the significance of this decrease should be specified.

Line 294, "Figure 4E" should be Figure 6E.

Line 300 If this is one of the main differences in brain specific serf knockout mice one would likely need blinded testing to prove it. At the very least, a large number of more examples would need to be in the supplement, in the case that quantifying this is technically difficult. One way could be to quantify the percent area that the brightest X% pixels occupy vs the percent area that the middle-grey pixels occupy. In the top-left, a circle containing the center of this feature would contain a higher percent area of the white pixels than the same circle in the top-right image.

Line 501 "E13.5 embryos" is probably a mistake here.

Figure 2A, This looks like much more of a difference in lag phase (the ability to recover) than a difference in growth rate. More details of how cells were treated before plating, are needed as this might affect lag time. The verbiage about differences in g
Line 568 Authors reference a detection algorithm in the results section, and methods include mention of the use of the ImageJ/FIJI software. However, authors do not include this detection algorithm in methods. Is this a plugin of ImageJ/FIJI? Did authors develop their own code? Is there some other software which they used to detect and quantify aggregates?

In Figure 5, the authors claim that there is difference in A β plaque formation between AM and AM;Serf2br^{-/-} mice. If any, the difference is rather subtle. How do the authors explain why there is significant difference using 6E10 but not W0-2 antibody in Figure 5F and 5G right panels? With the disperse data points, how accurate the fitting/trendline they depict in the correlation plot? Can the different "slope" they emphasized in Figure 5H be due to AM;Serf2br^{-/-} mice forms larger deposit with higher A β content?

In addition, statement in line 252-254 about Figure 5, "Some low level intracellular W0-2 and 6E10 staining could also be observed in the other 1-month-old AM mice, but not to the extent that it was picked up by the detection algorithm (Supplemental Figure 6A-B)." is confusing. Does it mean the detection algorithm is not accurate enough to reflect the staining? If so, how reliable these data extracted by this detection algorithm?

Correlation plots made in Figure 5 refer to "detailed stats in supp table 3". I still don't know what correlation method are used throughout the paper. This should be specified somewhere Supp table 3 is a table of pup counts meant for a different figure.

In Figure 6, the authors suggest the amyloid "plaques from AM;Serf2br^{-/-} mice tended to show more condensed plaques composed of short, thick and densely packed bundles of fibers with little space in between (Figure 6F, right panels)". This aligns with their finding that these plaques have "qFTAA-positive core, but virtually no hFTAA-staining". As the references they cited (Nystrom et al 2013 and Psonka-Antonczyk et al 2016) suggest, qFTAA staining indicates mature fiber formation and "coinciding with ThT and Congo red staining". Does it mean in the absence of SERF2, A β amyloid forms more mature fibers in AM mice? If yes, would this be contradictory to the previous findings of SERF proteins that they accelerate amyloid formation, shown by ThT aggregation assays? More discussion is needed.

Intensity ratio of LCO staining in Figure6 is 502nm/588nm, whereas the reference Nystrom et al 2013 used 500nm/540nm. Did the authors validate this change of wavelength for the assay?

Figure 6B is better to use bar graph with individual data points shown to show the distribution of the raw data.

According to Radde et al 2006, the 8-month-old APPPS1 mice has ~5 times more amyloid load in neocortex than 3-month-old mice. In addition, 8-month-old APPPS1 mice shows more amyloid-associated neuroinflammation and pathology. What's the rationale behind only examining 1-month-old and 3-month-old APPPS1 mice with or without SERF2?

As the authors point out, the CREbr^{-/-} causes decrease in brain size/weight. Did the authors carry out the phenotyping experiments in Figure 4 using CRE br^{-/-} mice as the control?

Line 703, Reference 29 and Reference 34 is identical.

Supp table It is good that the supp table with -log₂FC genes, as it clearly lists the FDR p-value, even though in the text they say all p-values that were multiple-testing-corrected were labeled Pbon. These aren't Pbon, of course, but it would be comforting to see the term FDR used in the main text when describing significance instead of "p-value".

Referee Cross-Comments

I agree with the comments made by reviewer #1 regarding the two obvious experiments that are lacking. In regards to the reviewer's statement that the characterizations appear to be appropriately done, overall I agree, but I have a lot of detailed comments as listed above.

Reviewer #1 (Comments to the Authors (Required)):

This manuscript presents a detailed analysis of SERF2 total and conditional knockout mice, and an analysis of how loss of this protein affects the well-studied APP-PS1 mouse. The characterizations appear to be appropriately done, and a suite of dye staining approaches have been used to quantify the (rather subtle) changes in plaque morphology and fibril structure that occur in the SERF2 conditional knockout background.

However, what is most striking is the absence of two seemingly obvious experiments:

- 1. using immunohistochemistry to assay the distribution of SERF2 in the APP/PS1 background (e.g., is SERF2 associated with intracellular Abeta or extracellular plaques?), There are multiple commercial sources of SERF2 antibodies,**

We agree with the reviewer that the knowing the distribution of SERF2 will be important to understand the mechanism by which its depletion changes the structure of aggregates. We did not include those data because in previous attempts, using antibodies that worked for Western blots, we have not been able to distinguish a signal in Serf2 +/+ brains over background staining in Serf2 -/- brains. For this revision, we have tried to (re)establish staining conditions for four different commercial and custom-made Serf2 antibodies, namely: i) Proteintech 11691 1 AP, ii) Biorad SERF2 antibody 34646 (custom made), iii) Sigma Aldrich SAB1100982, and iv) Novus biologicals NBP1 78375. In tests using DAB staining, only the NBP1 78375 antibody initially looked promising when comparing a Serf2-/- brain to a Serf2 +/+ brain. We then used this antibody and the same conditions for co-staining with amyloid beta with fluorescence labels (on 1 month and 3 month old brains, 3 brains per age-group, verified by genotype) and unfortunately, could not get any consistent or conclusive results, e.g. some Serf +/+ brains showed no staining whereas some Serf -/- brains showed abundant signal with the SERF2 antibody. From this we concluded that the Novus antibody was also not specific or sensitive enough to pick up a possible SERF-specific signal over the background binding in mouse brains. We therefore added the following sentence to the discussion:

“Future studies, including those assessing the association between SERF2 and amyloid beta in brain, will be required to determine mechanism by which depletion of Serf2 altered the amyloid structures.”

- 2. It is also surprising that no cognitive tests were done on the SERF2 KO/APP-PS1 mice, as this is the standard readout to determine if alterations in the APP- PS1 mice have AD-relevant outcomes. It is unclear to me why a paper with 25 authors did not attempt these experiments,**

We did breed cohorts for cognitive testing of amyloid-beta expressing mice in the past years but two independent attempts had to be terminated prematurely for different reasons. The first attempt with a commercial party could not be carried out anymore due to an infection in their animal facility and temporary closure of their phenotyping unit. The expanded mouse colony had to be terminated. During the second attempt at the German Mouse Clinic (of which phenotyping of the mice without amyloid beta has been successful and is described in manuscript) part of the amyloid-beta expressing mice developed seizures and died before phenotyping at the selected timepoint could take place. All remaining amyloid-beta mice had to be terminated for animal welfare reasons.

To note, both phenotyping attempts were planned with the ‘Borchelt’ mice as both parties had experience with exactly that model in their tests. For these mice, various degrees of seizures and premature death have been reported depending on their genetic background. We have not observed any problems while breeding these mice when they were still relatively young. We also never observed seizures and premature death in the Serf2 knockout mice or the ‘Radde’ APP-PS1 model used in this manuscript.

Given the mild effect of brain depletion of Serf2 on amyloid-beta aggregation, as also pointed out by the reviewer, and given the additional and unexpected phenotypes in the genetic background we used, we will not start a new attempt using a similar set up. To establish SERFs role in neurodegenerative-disease models other strategies will be required, for example by RNA-depletion of SERF2 only in adult brain. As proposed by referee #2, we do not want to further delay publishing our results, because they can be used to redirect efforts to establish the biological functions of SERF proteins, before establishing their function in mouse models for disease, which would still be very interesting.

Reviewer #2 (Comments to the Authors (Required)):

Detailed comments:

1. The title:

"SERF deletion modifies amyloid aggregation in a mouse model of Alzheimer's disease" might be made less declarative because of concerns expressed below.

“Deletion of SERF2 in mice delays embryonic development and alters amyloid deposit structure in brain”

2. Line 50 the word "show" is probably too strong here as different binding by structure specific amyloid dyes really just provides a preliminary indication that the structure of amyloid fibrils are different.

We agree with the referee and have changed the sentence to:

"In a mouse model for amyloid-beta aggregation, brain depletion of Serf2 altered the binding of structure-specific amyloid dyes, previously used to distinguish amyloid polymorphisms in human brain. These results suggest that Serf2 depletion changed the structure of amyloid deposits....."

3. Line 51-52 Blinded studies revealing statistically significant differences in fiber morphology would be far more convincing that the two images of fiber plaques shown, which are not convincing at all. What is needed would be either stats based on blind observers, or a lot of examples in the supplement plus a much more couched statement here. Something like "TEM imaging suggests that the structure of amyloid deposits could be altered by SERF2 knock-out, and further study would be needed to confirm this observation." The authors show just one example. This is definitely not enough to be remotely convincing given how easy it is to bias TEM data.

We agree with the referee that two images are not sufficient to state a difference in structure. From the current images it is not possible to establish the differences in fiber morphology quantitatively. We therefore follow the suggestion of the reviewer and change our statement to:

"which was further supported by scanning transmission electron microscopy but further study will be required to confirm this observation. "

4. Line 53 no aging studies were done so the words "and aging" should be deleted.

We removed "and aging"

5. Line 83-84, these lines are more appropriately nuanced and less declarative than the title, the title should be revised accordingly.

We agree with the suggestion of the reviewer and as mentioned in response to comment #1, propose the following title:

"Deletion of SERF2 in mice delays embryonic development and alters amyloid deposit structure in brain"

6. Line 86 That "Full-body Serf2 KO results in developmental delay and perinatal lethality in mice" is the strongest finding of the paper, but is not mentioned in the abstract.

We agree that this finding is important, even though it has also previously been described to be dependent on the genetic background in which Serf2 is deleted. In mixed genetic backgrounds Serf2 depleted mice are viable (see Cleverley et al. 2021), which is in line with our observations described in Supplementary Table 1. We, therefore, now added the following sentence to the abstract:

“Here, we generated conditional Serf2 knockout mice and found that full-body knockout of Serf2 delayed embryonic development, causing premature birth and perinatal lethality.”

7. Line 101 This is unclear and needs to be clarified. I don't understand how it is justified to lump two different values and compare them to one, an explicit comparison between +/- and +/+ to back up the 50% claim is necessary, is this decrease statistically significant? it seems like there is the claim in the figure with the 3 stars designation but is this to the wt or to the KO? Adding the lines between all the groups being compared in the graph would make this much clearer. It appears like the authors did what was appropriate, mathwise: ANOVA for all groups, then a posthoc test for individual comparisons.

Their text here...

(One-way ANOVA: Serf2+/- and Serf2-/- vs Serf2+/+)

...would be read/interpreted like this:

One-way ANOVA was used to compare the mean mRNA levels within all genotypes to look for the presence of any differences. A posthoc test revealed statistically significant differences specifically between both the Serf2+/+ vs Serf2 +/- and between Serf2+/+ and Serf2 -/-.

The interpretation of the reviewer is correct. A One-way ANOVA was indeed performed between all genotypes for both SERF1 and SERF2 mRNA measurements and a bonferroni corrected post-hoc analysis was used to reveal specific differences between genotype groups within this comparison. This was written in short hand to conserve space. We agree that adding lines to the graph could make this comparison more clear to the reader. These lines have therefore been added and changed the text accordingly

8. Line 102 is similarly brief and needs to be explicitly clarified

To clarify this, we added the P=value to the text and an “ns” indication in figure 1C.

9. Loss of serf2 (whole body and brain) resulted in a smaller mass of the whole body or the brain, respectively

To show that the loss of serf2 resulted in reduced embryo size, authors included measurements of body length (Fig1E) and body mass (SuppFig1E). While there is a significant different at E17.5 for the length measurements and not body mass (though body mass becomes significantly different between serf -/- and both serf+/- serf +/+ at P0), the differences in body mass on P0 is more convincing than a difference in body length at E17.5

We agree with the reviewer that the body mass difference at P0 is significant, which is why we felt it was important to include this data with the manuscript. However, we felt the body length measurements were a more complete and sensitive measurement, that also provided additional valuable insights into the growth trend at embryonic stages prior to P0. Therefore

we chose to include these in the main figure. Statistically speaking the difference in body mass and length are equally convincing. For length 2-way Anova $p_{\text{age}} < 0.0001$; $p_{\text{genotype}} < 0.0001$; $p_{\text{interaction}} = 0.0121$. For body mass 2-way Anova $p_{\text{age}} < 0.0001$; $p_{\text{genotype}} < 0.0001$; $p_{\text{interaction}} = 0.0262$.

As requested in the subsequent comments, we have added exact p-values to text and additional significance markers to the graphs to further clarify exact differences between the different genotypes.

10. Line 107 The statistics for E15.5 are missing from the graph, these should be done and shown. If the authors did the ANOVA for E15.5, the simplest thing is to add a bar with "ns" for not significant. Importantly the authors don't ever seem to add an "ns" bar anywhere in any of the figures, and while that does keep the graphs cleaner, it can just as easily be interpreted as a lack of significance testing.

As mentioned above, we did do the 2-way ANOVA for age and genotype with post-hoc analysis, including the E15.5 group, and choose to only depict the significant values in the graph for clarity. We have now added the more extensive analysis comparing all genotypes at every age and included both the significant and non-significant markings in the graph as requested.

11. Line 108 The actual p-values should be listed throughout the manuscript not just < values. This is particularly important for interpretation when the p-value is near alpha (0.05 in this case). For the same reason, "not significant/ns" should also be an actual value. That gives the reader the ability to see that 0.051 and 0.049 are actually quite close instead of ns vs *.

We agree with the reviewer that exact p-values should be provided to allow the reader to perform his/her own well-founded interpretation of the data, specifically for p-values near the alpha. We have therefore endeavored to provide these values throughout the text. The only exception being p-values that were > 0.999 or < 0.0001 , because we believe these extreme p-values are so distant from the alpha value that they leave no doubt in interpretation. The specific values mentioned in line 108 were a remnant of an initial analysis with a previous version of the statistical software that did not provide the exact p-values. We have re-run the analysis and included the exact p-values as requested. We hope to have addressed all values of concern in this new draft.

12. Line 109 "embryos during the embryonic stage leading up to birth (P0)" needs to be clarified, I think that what is meant is that the size differences were not apparent at any tested embryonic stage, including P0, for hets and homoz positive genotypes.

We thank the reviewer for notifying this and changed the sentence to:

"No differences in size could be observed between homozygous wild type Serf2^{+/+}, and heterozygous Serf2^{+/-} embryos at any embryonic stage, including P0 (Two-way ANOVA for both body length and weight: at all ages $p_{\text{bon}} > 0.9999$)."

13. Line 111 cause is misspelt

We changed this to “cause”.

14. Line 113 "indicates" is too strong, "suggests" is a better choice

We agree and changed “indicates” to “suggests”

15. Line 122 the phenotypes if any of serf+/- pups should be disclosed and discussed.

Serf+/- pups were born and survived at expected numbers and showed no growth delay, resembling the phenotypes of the Serf +/+ pups (Figures 1B and 1E). With respect to the phenotype of partial atelectasis, in one test cohort of 30 pups, which included 20 Serf +/- pups, only one Serf +/- pup showed partial fetal atelectasis at birth as opposed to 7 out of 7 of the Serf -/- pups and 0 out of 3 Serf +/+ pups. Higher numbers, in particular of the Serf +/- pups, will be required to determine whether the Serf +/- pups have an increased incidence of fetal atelectasis compared to Serf +/+ pups.

16. The rationale in Lines 123-134 and Figure 2 is not clear nor convincing to explain the phenotypes they observed in cell line knockout and mouse knockout. With the upregulation of cell division factors and downregulation of growth arrest factors in serf2-/- MEFs shown by RNA-seq, wouldn't the authors expect to see faster cell division in these serf2-/- MEFs compared to WT MEFs? How does this RNA-seq data explain the slower proliferation they observed in serf2-/- HEK293T cells?

We understand the confusion of the referee and now clarified our statement in the text as follows.

“The upregulation of cell cycle regulators may seem counterintuitive in cells that divide slower. However, the upregulation of cell cycle factor does not automatically mean that cell divide faster. It could be that they need longer time to passage through S- or G2 because they experience replication stress and need to repair DNA in G2 phase. This leads to upregulation of cell cycle factors as well, because they need longer time to passage through these phases.”

17. Line 125 the words cell intrinsic are misleading, I think what is meant is cell autonomous.

Although they show that Serf2-/- HEK293T knockout cell lines have reduced proliferation (Fig2A) similar results have been shown before, they did reference van Ham 2010, which showed that serf/moag-4 acted in a cell autonomous manner, but authors did not reference Tsuboyama 2020 which also used CRISPR/Cas9 to knockout serf2 in HEK293T cells, and showed a growth defect, this paper needs to be cited.

We agree with the referee and have changed cell intrinsic to cell autonomous. We have now also referred to Tsuboyama et al., 2020.

18. Line 129 I found this rather superficial analysis unfortunate. One would hope that

authors would follow where data leads, overall impression that they were more intent on proving a connection to amyloidogenesis than following the new observations in their data.

We agree with the referee that the embryonic phenotype deserves attention. We followed up on it to the extent we did, because the primary question we aimed to address in this study was related to the effect of SERF deletion on disease-protein aggregation in mouse brain. The referee points at a strong phenotype of lung development, but we would like to clarify that this phenotype is likely due of premature birth and not caused by a specific function of Serf in lung, rather explained by slow growth as a consequence of Serf depletion. We do find the endogenous role of Serf in cell growth and development identified in this study exciting. Such role was not apparent from our previous *C. elegans* studies and may explain why a protein that enhances the toxicity of aggregation-prone proteins has been conserved in evolution. We agree that this would be a productive direction to follow in future studies in cell- and animal models, but we will not follow up on this in mice, because this is beyond our expertise.

19. Line 137-139 I found this rational and transition rather strange, it appears that the authors are more intent on accumulating data in favor of their initial hypothesis, that SERF is involved in amyloid formation, rather than following the clearest effects of SERF knockouts which seem to be on development.

As mentioned previously, the perinatal death has previously been reported by Cleverley et al 2021 and ascribed to the genetic background in which SERF2 was depleted. Our results suggest that this is due to a general delay in cell growth during embryonic development and premature birth. Diving into the endogenous function of SERF in cell growth, e.g. by following up on the transcriptional profile of the Serf2 deletion cells, will require an independent and detailed molecular study, which was beyond the aim of this study. We removed sentences 137-139 and changed sentences 141-142 from the original manuscript to:

“Going back to our initial aim to establish the effect of SERF2 depletion on amyloid aggregation in mouse brain we needed to circumvent the perinatal lethality of full-body Serf2 KO mice.”

20. Line 154 This is the first time I've ever seen an interaction term from an ANOVA used correctly in paper, nice!

We thank the referee for this compliment.

21. Line 161-165 It is unclear if the cre effect is sufficient to explain the whole effect or not, until this is explicitly shown is not the references to reduced weight should be deleted. Because of the way the authors split up these data (WT vs Cre, then WT vs Serf2) it probably means they did the experiments at different times and cannot compare directly. Alternatively one might be able compare normalized data (normalized differences from WT at each age for each tested genotype could be used in a hypothesis test between genotypes to answer that.

Indeed these experiments were done at different times. Instead of normalizing the data, which would still leave room for experiment-specific biological variation, we propose to change the sentence to:

“Therefore, it is possible that the expression of Cre alone contributed to the observed decrease in brain size.”

22. Figure 4 I'm a little surprised the authors don't refer to Pbon for these. Or some other multiple-testing-correction.

This is indeed a valid remark by the reviewer. Given that this analysis deals with a large number of different experiments comparing a varying number of unique parameters between only two test groups we felt a bonferoni correction would not be appropriate and to stringent. We did debate using a more stringent significance cut-off or FDR procedure, but in the end decided against this. A more stringent significance cut-off would be fairly arbitrary and FDR is usually used for correcting multiple tests of the same nature. The tests in Figure 4 are all analyzed separately through varying statistical models, appropriate for the individual behavioral test and their associated confounding factors. We therefore felt that selecting one general, appropriate statistical correction was not really possible and a wrong selection might misrepresent the data.

Instead we chose to display uncorrected data and interpret the individual data from the tests in the larger behavioral context. Usually, one might perform only one or two behavioral tests and this would not be possible. Here, we chose to do a broad characterization. This allows us to look at a significant parameter not only in the context of the other parameters within that test, but also in the context of other tests evaluating the same modalities. As discussed in the text, by looking at the size of the observed effects and the consistency across parameters and tests, we can identify general behavioral patterns and those single, inconsistent significant measurements that are most likely false positive measurements. We believe this provides us with the most sound analysis of this highly varied data set and it gives readers the best opportunity to interpret the data for themselves and compare it with their own outcomes.

23. Line 185-87. The authors seem appropriately cautious in the interpretation of the behavioral data

We thank the referee for this positive comment

24. Line 188 "proceed with this model" What model exactly?

To clarify this, we changed the text to “...these brain-specific knockout mice....”

25. Line 242 the slight increase in plaque amounts is really slight. Yes, Pbon is over 0.05, but, it is quite common to hold the multiple-testing-correction p's to a different standard than the initial p-values (for too many reasons for this to list here). A way to deal with this

in a paper is to say our alpha for hypothesis testing was 0.05, and our alpha after multiple-testing-correction was 0.1. (Which is routine in bioinf papers)

The reviewer is correct that the increase in plaque amount is slight and does not reach the significance threshold of alpha 0.05, which we use throughout the manuscript for both hypothesis and multiple correction testing. While it is correct that bioinformatics papers often choose a different alpha for multiple corrections, the amount of comparisons being corrected for in these papers is usually also significantly higher. We therefore chose to use a more stringent approach by using the bonferroni correction and 0.05 alpha. It is true that this approach may result in certain differences just failing to reach significance. However, we opted to err on the side of caution and perhaps be too stringent, limiting the type 1 error rate. As the reviewer indicates below, the selection of these limits is indeed arbitrary. This why in our manuscript we also discuss the trends in these borderline values and provide the reader with the exact p-values to allow them to make their own evaluation of significance.

Moreover, in the case of differences in absolute plaque amounts in our manuscript we also see that the high variability in amyloid-beta and plaque load between animals, plays a role in the power of our statistical tests to detect these differences, which is why we included the linear regression analysis taking into account this variability.

26. Line 247 Is this small increase statistically significant? See prev comment. "Statistically (in)significant" is a prior agreement with the reader on an arbitrary alpha value. The standard is 0.05 for a hypothesis test, and but is not standardized for multiple-testing-corrections. They need to spell out precisely what they're threshold is.

The alpha threshold for all statistical testing is 0.05. This has now been clearly specified in the materials and methods section for the statistical analysis. The reason for this threshold is discussed above. As stated above, we invite the reader to make their own evaluation about the biological significance of these findings based on the data presented.

27. Line 294, "Figure 4E" should be Figure 6E.

We changed Figure 4E for Figure 6E in the revised text

28. Line 300 If this is one of the main differences in brain specific serf knockout mice one would likely need blinded testing to prove it. At the very least, a large number of more examples would need to be in the supplement, in the case that quantifying this is technically difficult.

One way could be to quantify the percent area that the brightest X% pixels occupy vs the percent area that the middle-grey pixels occupy. In the top-left, a circle containing the center of this feature would contain a higher percent area of the white pixels than the same circle in the top-right image.

As mentioned above, we agree with the referee that two images are not sufficient to state a difference in structure. From the current images it is not possible to establish the differences in fiber morphology quantitatively. The EM images we included were two out of 8 or 9 for each condition. These images were blinded and sorted by an EM microscopist, which matched the genotypes, but the number of images were not sufficient to make the analysis statistically solid. We added the other images in the source data of Figure 6 and follow the previous suggestion of the reviewer by changing our statement in the abstract to:

“These results suggest that Serf2 depletion changed the structure of amyloid deposits, which was further supported by scanning transmission electron microscopy but further study will be required to confirm this observation.”

29. Line 501 "E13.5 embryos" is probably a mistake here.

If the referee means to refer to the section cell culture, then no this is not a mistake, this is the stage at which the MEFs were isolated.

30. Figure 2A, This looks like much more of a difference in lag phase (the ability to recover) than a difference in growth rate. More details of how cells were treated before plating, are needed as this might affect lag time. The verbiage about differences in g

We agree with the reviewer that the observed difference in growth rate is not necessarily due to more time spent in cell division or specific cell division phases. As mentioned in remark 15, it is possible that the delay in growth may be due to cells being less robust, stress resistant, or prone to errors and damage during cell division, resulting in more time spent in recovery before or between cell division. This would be in line with an extended lag phase. The exact nature of the delay cell growth and how SERF2 contributes to the prevention of this, is certainly an interesting avenue for further research into the endogenous role of SERF2.

31. Line 568 Authors reference a detection algorithm in the results section, and methods include mention of the use of the ImageJ/FIJI software. However, authors do not include this detection algorithm in methods. Is this a plugin of ImageJ/FIJI? Did authors develop their own code? Is there some other software which they used to detect and quantify aggregates?

The M&M sections specifies the “Triangle dark” threshold algorithm of the Fiji Software that was used, as well as the cut-off values used for plaque detection.

32. In Figure 5, the authors claim that there is difference in A β plaque formation between AM and AM;Serf2br^{-/-} mice. If any, the difference is rather subtle. How do the authors explain why there is significant difference using 6E10 but not W0-2 antibody in Figure 5F and 5G right panels? With the disperse data points, how accurate the fitting/trendline

they depict in the correlation plot? Can the different "slope" they emphasized in Figure 5H be due to AM;Serf2^{br/-} mice forms larger deposit with higher A β content?

The difference is indeed subtle and differences are difficult to detect because of the large intra-individual variability between mice. This is why we included the correlation analysis to illustrate this. We do not want to overstate this difference. The goal is again to present the reader with the available data and show that in addition to the structural differences we show in figure 6, some subtle differences may also be detected in these quantitative measures, but potentially masked by the large variability. For this reason we also include the fit parameters of the correlation in the supplemental data and invite the reader to make their own evaluation of the importance of the data. As for the difference between the 6E10 and W0-2 staining, we have no definitive explanation for this. These are two distinct antibodies that partially share the same epitope binding, but are not identical. As we believe the differences seen are due to structural changes, we believe it is not unlikely that there is a small difference in binding affinity. However, as mentioned previously, the high degree in variation could also be a factor and contribute to a lack of power. The difference in slope in 5H, can definitely be caused by a difference in plaque composition. In terms of size, we detected a lot of variation in size of plaques in both groups (see suppl fig 3B-D), but did not detect a significant difference between groups in general morphological parameters. The exact Abeta content of an individual plaque is of course impossible to gauge. We do see more dense, mature plaques in the SERF2 KO group, so a higher Abeta content in these plaques would be likely and could indeed offer an explanation for the difference seen.

33. In addition, statement in line 252-254 about Figure 5, "Some low level intracellular W0-2 and 6E10 staining could also be observed in the other 1-month-old AM mice, not to the extent that it was picked up by the detection algorithm (Supplemental Figure 6A-B)." is confusing. Does it mean the detection algorithm is not accurate enough to reflect the staining? If so, how reliable these data extracted by this detection algorithm?

The detection algorithm was made to detect plaques using an initial test set of images and compared to manual counting for sensitivity and specificity. For detection, it depends on parameters based on size and signal intensity versus background. It was not developed to detect intracellular staining. In the pre-plaque 1 month group we identified an unexpected and unusual increase in staining being detected and upon closer examination we found that this was due to increased intracellular staining in this group. So for the detection of plaques this algorithm was tested and was very accurate to discriminate plaques from background staining in most strains. Just for select cases within the AM;Serf2^{br/-} with increased intracellular amyloid beta staining substantially above the background, the algorithm detected and counted this signal as well. We were not looking for this feature and, as mentioned in the manuscript, this increase did not amount to a statistically significant difference with the other strains. However, we felt that given the noticeable trend it was a potentially relevant scientific finding and worth addressing and sharing with the reader.

34. Correlation plots made in Figure 5 refer to "detailed stats in supp table 3". I still don't

know what correlation method are used throughout the paper. This should be specified somewhere Supp table 3 is a table of pup counts meant for a different figure.

For the correlation plots the simple linear regression of the graphpad software was used. This is now also specified in the statistical section of the M&M. The correct table ref is supplemental table 4. We have changed this accordingly in the text.

35. In Figure 6, the authors suggest the amyloid "plaques from AM;Serf2br-/- mice tended to show more condensed plaques composed of short, thick and densely packed bundles of fibers with little space in between (Figure 6F, right panels)". This aligns with their finding that these plaques have "qFTAA-positive core, but virtually no hFTAA-staining". As the references they cited (Nystrom et al 2013 and Psonka-Antonczyk et al 2016) suggest, qFTAA staining indicates mature fiber formation and "coinciding with ThT and Congo red staining". Does it mean in the absence of SERF2, A β amyloid forms more mature fibers in AM mice? If yes, would this be contradictory to the previous findings of SERF proteins that they accelerate amyloid formation, shown by ThT aggregation assays? More discussion is needed.

We agree with the referee that based on the cited references, the staining pattern of the amyloid deposits in the Serfs deletion brains would resemble more mature plaques. If so, this would seemingly contradict findings in previous studies, such as the in vitro aggregation results in Pras et al., 2021. However, as mentioned in the discussion, in these cell free experiments, with a finite amount of aggregating protein, SERF2 mainly acts on nucleation. It is therefore not illogical to hypothesize that the removal of Serf2 could shift the balance in the aggregation kinetics towards elongation and maturation instead of the formation of new nuclei, resulting in more mature fibrils. Based on our current data, we cannot say if the observed structural differences are simply the consequence of a temporal shift in the formation of mature fibrils, a structural difference in the nuclei and resulting fibrils being formed or a combination of both. To answer this question will require further in-depth temporal and structural analyses. In an attempt to further clarify and address the reviewers concerns, we have added the following to our discussion:

"Based on previous studies, the LCO spectra observed in brains of Serf2-deletion mice, however, would resemble spectra of increased mature fibrils^{42,43}. If so, this would seemingly contradict findings in previous studies, such as the in vitro aggregation results in Pras et al., 2021. However, with a finite amount of aggregating protein, SERF2 mainly acts on nucleation. It is therefore possible that removal of Serf2 could shift the balance in the aggregation kinetics towards elongation and maturation instead of the formation of new nuclei, resulting in more mature fibrils. Further in-depth temporal and structural analyses, including those assessing associations between SERF2 and amyloid beta in brain, will be required to determine mechanism by which depletion of Serf2 altered the amyloid structures."

36. Intensity ratio of LCO staining in Figure6 is 502nm/588nm, whereas the reference

Nystrom et al 2013 used 500nm/540nm. Did the authors validate this change of wavelength for the assay?

These experiments were performed in the lab of Mathias Jucker. While the original source publication for the use of these LCOs to characterize plaque structure is the publication by Nilson, the Jucker lab has previously verified in other mouse models the two-peak spectrum of hFTAA (with peaks around 550nm and 590nm). They also confirmed a spectrum with a peak around 500nm for qFTAA with a long taper toward to higher (red) wavelengths (as illustrated in figure 6A). By selecting the higher wavelength 588nm peak, we reduced bleed through from the qFTAA stain in the red channel. We have added the following references to the manuscript in which the use of these wavelengths is described:

Rasmussen, J. et al. Amyloid polymorphisms constitute distinct clouds of conformational variants in different etiological subtypes of Alzheimer's disease. Proceedings of the National Academy of Sciences 114, 13018–13023 (2017).

37. Figure 6B is better to use bar graph with individual data points shown to show the distribution of the raw data.

The individual datapoints are depicted in figure C, figure B was meant to show that the data are distributed in three different intensity groups.

38. According to Radde et al 2006, the 8-month-old APPPS1 mice has ~5 times more amyloid load in neocortex than 3-month-old mice. In addition, 8-month-old APPPS1 mice shows more amyloid-associated neuroinflammation and pathology. What's the rationale behind only examining 1-month-old and 3-month-old APPPS1 mice with or without SERF2?

Our study using the Radde was designed to look at the effect of SERF depletion on aggregation. We only examined 1 month and 3 month mice, because of previous studies showing that MOAG-4 and SERF accelerate the formation of seeds that catalyze the formation of amyloid, which delays aggregation but at some point reaches a plateau stage. We reasoned that if deletion of SERF would result in a difference in aggregation, this should be visible in early stages.

39. As the authors point out, the CREbr^{-/-} causes decrease in brain size/weight. Did the authors carry out the phenotyping experiments in Figure 4 using CRE br^{-/-} mice as the control?

We initially planned to include these mice in our phenotypic analyses as well, but could not do so, because of funding and capacity limitations at the GMC. We therefore prioritized the Serf2 brain deletion mice and their controls either with or without amyloid beta, (of which the amyloid-beta mice had to be terminated before phenotyping see also our response to comment #2). Given the minimal phenotypic effects of Serf2 deletion on behavioral

phenotypes (Figure 4 and description), which is in the CREbr^{-/-} background, we do not expect that phenotyping CREbr^{-/-} mice alone will reveal additional information.

40. Line 703, Reference 29 and Reference 34 is identical.

We thank the reviewer for noticing this and we corrected this mistake

41. Supp table It is good that the supp table with $-\log_2FC$ genes, as it clearly lists the FDR p-value, even though in the text they say all p-values that were multiple-testing-corrected were labeled Pbon. These aren't Pbon, of course, but it would be comforting to see the term FDR used in the main text when describing significance instead of "p-value".

The use of FDR for the analysis of the RNA sequencing data is documented in the "RNA sequencing" section of the M&M in the main text. We have also included a reference to this detailed description in the statistical section of the M&M to clarify the use of different statistical methods and software for this analysis.

42. Referee Cross-Comments

I agree with the comments made by reviewer #1 regarding the two obvious experiments that are lacking. In regards to the reviewer's statement that the characterizations appear to be appropriately done, overall I agree, but I have a lot of detailed comments as listed above.

See our responses #1 and #2 to reviewer #1

April 11, 2023

RE: Life Science Alliance Manuscript #LSA-2022-01730-TR

Dr. Ellen AA Nollen
University Medical Center Groningen
European Research Institute for the Biology of Ageing (Eriba)
PO Box 196
Groningen 9700AD
Netherlands

Dear Dr. Nollen,

Thank you for submitting your revised manuscript entitled "Deletion of SERF2 in mice delays embryonic development and alters amyloid deposit structure in brain". We would be happy to publish your paper in Life Science Alliance pending final revisions necessary to meet our formatting guidelines.

- please upload your table files as editable doc or excel files
- please add ORCID ID for secondary corresponding author-you should have received instructions on how to do so
- please make sure that the author order entered in our system and the author order in the manuscript match and that every author is entered in our system
- please use the [10 author names, et al.] format in your references (i.e. limit the author names to the first 10)
- please add a callout For Figure S1B,C; Figure S4; and Figure S5E-F to your main manuscript text

A. FINAL FILES:

B. MANUSCRIPT ORGANIZATION AND FORMATTING:

Sincerely,

April 18, 2023

RE: Life Science Alliance Manuscript #LSA-2022-01730-TRR

Dr. Ellen AA Nollen
University Medical Center Groningen
European Research Institute for the Biology of Ageing (Eriba)
PO Box 196
Groningen 9700AD
Netherlands

Dear Dr. Nollen,

Thank you for submitting your Research Article entitled "Deletion of SERF2 in mice delays embryonic development and alters amyloid deposit structure in brain". It is a pleasure to let you know that your manuscript is now accepted for publication in Life Science Alliance. Congratulations on this interesting work.

DISTRIBUTION OF MATERIALS:

Again, congratulations on a very nice paper. I hope you found the review process to be constructive and are pleased with how the manuscript was handled editorially. We look forward to future exciting submissions from your lab.

Sincerely,
